# EmbedDistill: A geometric knowledge distillation for information retrieval

## Abstract

Large neural models (such as Transformers) achieve state-of-the-art performance for information retrieval. In this paper, we aim to improve distillation methods that pave the way for the deployment of such models in practice. The proposed distillation approach supports both retrieval and re-ranking stages and crucially leverages the relative geometry among queries and documents learned by the large teacher model. It goes beyond existing distillation methods in the information retrieval literature, which simply rely on the teacher's scalar scores over the training data, on two fronts: providing stronger signals about local geometry via embedding matching and attaining better coverage of data manifold globally via query generation. Embedding matching provides a stronger signal to align the representations of the teacher and student models. At the same time, query generation explores the data manifold to reduce the discrepancies between the student and teacher where the training data is sparse. Our distillation approach is theoretically justified and applies to both dual encoder (DE) and cross-encoder (CE) models. Furthermore, for distilling a CE model to a DE model via embedding matching, we propose a novel dual pooling-based scorer for the CE model that facilitates a more distillation-friendly embedding geometry, especially for DE student models.

## 1 Introduction

Neural models for information retrieval (IR) are increasingly used to capture the true ranking in various applications, including web search (Mitra & Craswell, 2018), recommendation (Zhang et al., 2019), and question-answering (QA; Chen et al. 2017). Notably, the recent success of Transformers (Vaswani et al., 2017)-based pre-trained language models (Devlin et al., 2019; Liu et al., 2019; Raffel et al., 2020) on a wide range of natural language understanding tasks has prompted their utilization in IR to capture query-document relevance (see, e.g., Dai & Callan, 2019b; MacAvaney et al., 2019a; Nogueira & Cho, 2019; Lee et al., 2019; Karpukhin et al., 2020, and references therein).

A typical IR system comprises two stages: (1) A *retriever* first selects a small subset of potentially relevant candidate documents (out of a large collection) for a given query; and (2) A *re-ranker* then identifies a precise ranking among the candidates provided by the retriever. *Dual-encoder* (DE) models are the de-facto architecture for retrievers (Lee et al., 2019; Karpukhin et al., 2020) Such models independently embed queries and documents into a common space, and capture their relevance by simple operations on these embeddings such as the inner product. This enables offline creation of a document index and supports fast retrieval during inference via efficient maximum inner product search (MIPS) implementations (Johnson et al., 2021; Guo et al., 2020), with query embedding generation primarily dictating the inference latency. *Cross-encoder* (CE) models, on the other hand, are preferred as re-rankers, owing to their excellent performance (Nogueira & Cho, 2019; Dai & Callan, 2019a; Yilmaz et al., 2019). A CE model jointly encodes a query-document pair while enabling early interaction among query and document text. Employing a CE model for retrieval is often infeasible, as it would require processing a given query with *every* document in the collection at inference time. In fact, even in the re-ranking stage, the inference cost of CE models is high enough (Khattab & Zaharia, 2020) to warrant exploration of efficient alternatives (Hofstätter et al., 2020; Khattab & Zaharia, 2020; Menon et al., 2022).

*Knowledge distillation* (Bucilǎ et al., 2006; Hinton et al., 2015) provides a general strategy to address the prohibitive inference cost associated with high-quality large neural models. In the IR literature,

most existing distillation methods only rely on the teacher's query-document relevance scores (see, e.g., Chen et al., 2021; Lu et al., 2020; Hofstätter et al., 2020; Ren et al., 2021; Santhanam et al., 2021) or their proxies (Izacard & Grave, 2021). However, given that neural IR models are inherently embedding-based, it is natural to ask: is it useful to go beyond matching of the teacher and student models' *scores*, and directly aim to align their *embedding spaces*?

With this in mind, we propose a novel distillation method for IR models that utilizes an *embedding matching* task to train the student. The proposed method supports *cross-architecture distillation* and improves upon existing distillation methods for both retriever and re-ranker models. When distilling a large DE model into a smaller DE model, for a given query (document), one can minimize the distance between the query (document) embeddings of the teacher and student after compatible projection layers to account for any dimensionality mismatch. In contrast, defining an embedding matching task for distilling a CE model into a DE model is not as straightforward. For Transformers-based CE models, it is common to use the final embedding of a special token, e.g., `[CLS]` in BERT (Devlin et al., 2019), to compute query-document relevance (Nogueira & Cho, 2019). However, as we note in Sec. 4.2, this token embedding does not capture semantic similarity between the query and document. To make CE models more amenable to distillation via embedding matching, we propose a modified CE scoring approach by utilizing a novel *dual-pooling* strategy: this separately pools the final query and document token embeddings, and then computes the inner product between the pooled embeddings as the relevance score.

Our key contributions toward improving IR models via distillation are:

- We propose a novel distillation approach for neural IR models, namely EmbedDistill, that goes beyond score matching and aligns the embedding spaces of the teacher and student models.
- We consider a novel DE to DE distillation setup to showcase the effectiveness of our embedding matching approach (Sec. 4.1). Specifically, we consider a student DE model with an asymmetric configuration, consisting of a small query encoder and a *frozen* encoder inherited from the teacher. Such a configuration significantly reduces query embedding generation latency during inference, while leveraging the teachers' high-quality document index.
- We show that our proposed distillation approach can leverage synthetic data to improve student by further aligning the embedding spaces of the teacher and student (Sec. 4.3).
- We theoretically justify both embedding matching and query generation components of our proposed method (Sec. 5). Further, we provide a comprehensive empirical evaluation of the method (Sec. 6) on two standard IR benchmarks – Natural Questions (Kwiatkowski et al., 2019a) and MSMARCO (Nguyen et al., 2016).
- Finally, we demonstrate the utility of embedding matching for CE to DE distillation on MSMARCO by utilizing a novel pooling strategy for CE models, namely *dual pooling* (Sec. 4.2), which might be of independent interest.

## 2 RELATED WORK

Here, we review some existing Transformers-based IR models, and discuss prior work on distillation and data augmentation for such models. We also cover prior efforts on aligning representations during distillation for non-IR settings. Unlike our problem setting where the DE student is factorized, these works mainly consider distilling a single large Transformer into a smaller one.

**Transformers-based architectures for IR.** Besides DE and CE models described in Section 1, intermediate configurations (MacAvaney et al., 2020; Khattab & Zaharia, 2020; Nie et al., 2020; Luan et al., 2021) have been proposed. Such models first independently encode the query and document, and then apply a more complex *late interaction* between the two. Interestingly, Nogueira et al. (2020) explore generative encoder-decoder style model for re-ranking, where a T5 (Raffel et al., 2020) model takes a query-document pair as input and its score for certain target tokens (e.g., `True/False`) defines the relevance score for the pair. In this paper, we focus on basic DE and CE models to showcase the benefits of our proposed geometric distillation approach. Exploring embedding matching for the aforementioned architectures is an interesting avenue for future exploration.

**Distillation for IR.** Traditional distillation techniques have been widely applied in the IR literature, often to distill a teacher CE model to a student DE model (Chen et al., 2021; Li et al., 2020). Recently, distillation from a DE model (with complex late interaction) to another DE model (with

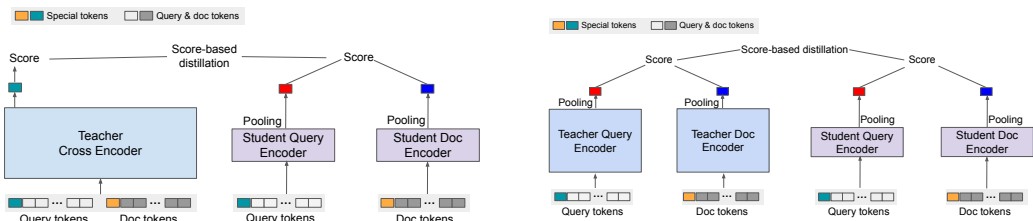

(a) Score-based CE to DE distillation         (b) Score-based DE to DE distillation

Figure 1: Illustration of score-based distillation for IR (cf. Section 3.2). Fig. 1a describes distillation from a teacher [CLS]-pooled CE model to a student DE model. Fig. 1b depicts distillation from a teacher DE model to a student DE model. Both distillation setup employ symmetric DE configurations where query and document encoders of the student model are of the same size.

inner-product scoring) has also been considered (Lin et al., 2021; Hofstätter et al., 2021). As for distilling across different model architectures, Lu et al. (2020); Izacard & Grave (2021) consider distillation from a teacher CE model to a student DE model. Hofstätter et al. (2020) conduct an extensive study of knowledge distillation across a wide-range of model architectures. Most existing distillation schemes for IR rely on only teacher scores; by contrast, we propose a geometric approach that also utilizes the teacher *embeddings*. Many recent efforts (Qu et al., 2021; Ren et al., 2021; Santhanam et al., 2021) show that iterative multi-stage (self-)distillation improves upon single-stage distillation (Qu et al., 2021; Ren et al., 2021; Santhanam et al., 2021). These approaches use a model from the previous stage to obtain labels (Santhanam et al., 2021) as well as mine harder-negatives (Xiong et al., 2021). We only focus on the single-stage training in this paper. Multi-stage procedures are complementary to our work, as one can employ our proposed embedding-matching approach in various stages of such a procedure.

**Distillation with representation alignments.** Outside of the IR context, a few prior works proposed to utilize alignment between hidden layers during distillation (Romero et al., 2014; Sanh et al., 2019; Jiao et al., 2020; Aguilar et al., 2020; Zhang & Ma, 2020). Chen et al. (2022) utilize the representation alignment to re-use teacher's classification layer. Our work differs from these as it needs to address multiple challenges presented by an IR setting: 1) cross-architecture distillation such as CE to DE distillation; 2) partial representation alignment of query or document representations as opposed to aligning representations for the entire input, i.e., a query-documents pair; 3) catering representation alignment approach to novel IR setups such as asymmetric DE configuration; and 4) dealing with negative sampling due to a large number of classes (documents). To the best of our knowledge, our work is the first in the IR literature that goes beyond simply matching scores (or its proxies).

**Semi-supervised learning for IR.** Data augmentation or semi-supervised learning has been previously used to ensure data efficiency in IR (see, e.g., Zhao et al., 2021; MacAvaney et al., 2019b). More interestingly, data augmentation via large pre-trained models have enabled performance improvements as well. Doc2query (Nogueira et al., 2019b;a) performs document expansion by generating queries that are relevant to the document and appending those queries to the document. Query expansion has also been considered, e.g., for document re-ranking (Zheng et al., 2020). Notably, generating synthetic (query, passage, answer) triples from a text corpus to augment existing training data for QA systems also leads to significant gains (Alberti et al., 2019; Oğuz et al., 2021). Furthermore, even zero-shot approaches, where no labeled query-document pairs are used, can also perform competitively to supervised methods. Such methods train a DE model by relying on inverse cloze task (Lee et al., 2019; Izacard et al., 2021), synthetic query-document pairs given a target text corpus (Ma et al., 2021), or relevance scores from large pretrained models (Sachan et al., 2022). Unlike these works, we utilize query-generation capability to ensure tighter alignment between the embedding spaces of the teacher and student.

## 3 BACKGROUND

Let $\mathcal{Q}$ and $\mathcal{D}$ denote the query and document spaces, respectively. An IR model is equivalent to a scorer $s : \mathcal{Q} \times \mathcal{D} \to \mathbb{R}$, i.e., it assigns a (relevance) score $s(q, d)$ for a query-document pair $(q, d) \in \mathcal{Q} \times \mathcal{D}$. Ideally, we want to learn an IR model or scorer that is faithful to the true query-document relevance, i.e., $s(q, d) > s(q, d')$ *iff* the document $d$ is more relevant to the query $q$ than document $d'$. We assume access to $n$ labeled training examples of the form $\mathcal{S}_n = \{(q_i, \mathbf{d}_i, \mathbf{y}_i)\}_{i \in [n]}$. Here,

$\mathbf{d}_i = (d_{i,1}, \ldots, d_{i,L}) \in \mathcal{D}^L, \ \forall i \in [n]$, denotes a list of $L$ documents and $\mathbf{y}_i = (y_{i,1}, \ldots, y_{i,L}) \in \{0,1\}^L$ denotes the corresponding labels such that $y_{i,j} = 1$ iff the document $d_{i,j}$ is relevant to the query $q_i$. Given $\mathcal{S}_n$, we learn an IR model by minimizing

$$R(s; \mathcal{S}_n) = \frac{1}{n} \sum\nolimits_{i \in [n]} \ell(s_{q_i, \mathbf{d}_i}, \mathbf{y}_i), \tag{1}$$

where $\ell(s_{q_i, \mathbf{d}_i} := (s(q_i, d_{1,i}), \ldots, s(q_i, d_{1,L})), \mathbf{y}_i)$ denotes the loss $s$ incurs on $(q_i, \mathbf{d}_i, \mathbf{y}_i)$. Due to space constraint, we present a few common choices for the loss function $\ell$ in Appendix A.

While this learning framework is general enough to work with any IR models as the scorers, next, we formally describe two families of IR models that are prevalent in the recent literature.

### 3.1 TRANSFORMER-BASED IR MODELS: CROSS-ENCODERS AND DUAL-ENCODERS

Let query $q = (q^1, \ldots, q^{m_1})$ and document $d = (d^1, \ldots, d^{m_2})$ consist of $m_1$ and $m_2$ tokens, respectively. We now discuss how Transformers-based CE and DE models process a given $(q, d)$ pair.

**Cross-encoder model.** Let $o = [q; d]$ denote the $m_1 + m_2$ length sequence obtained by concatenating $q$ and $d$. Furthermore, let $\tilde{o}$ be the sequence obtained by adding special tokens such $[\texttt{CLS}]$ and $[\texttt{SEP}]$ to $o$ at appropriate locations. Now, given an encoder-only Transformer model $\mathrm{Enc}$, the relevance score for the query-document pair $(q, d)$ is defined as

$$s(q, d) = \langle w, \mathrm{pool}(\mathrm{Enc}(\tilde{o})) \rangle = \langle w, \texttt{emb}_{q,d} \rangle, \tag{2}$$

where $w$ is a $d$-dimensional classification vector, and $\mathrm{pool}(\cdot)$ denotes a pooling operation that transform $\mathrm{Enc}(\tilde{o})$ — contextualized token embeddings produced by $\mathrm{Enc}$ — to a joint embedding vector $\texttt{emb}_{q,d}^{\mathrm{t}}$. $[\texttt{CLS}]$-pooling is a common strategy that simply outputs the embedding of the $[\texttt{CLS}]$ token.

**Dual-encoder model.** Let $\tilde{q}$ and $\tilde{d}$ be the sequences obtained by adding appropriate special tokens to $q$ and $d$, respectively. A DE model comprises two (encoder-only) Transformers $\mathrm{Enc}_Q$ and $\mathrm{Enc}_D$, which we call query and document encoders, respectively.[1] Let $\texttt{emb}_q = \mathrm{pool}(\mathrm{Enc}_Q(\tilde{q}))$ and $\texttt{emb}_d = \mathrm{pool}(\mathrm{Enc}_D(\tilde{d}))$ denote the query and document embeddings, respectively. Now, one can define $s(q, d) = \langle \mathrm{pool}(\texttt{emb}_q, \texttt{emb}_d \rangle$ to be the relevance score assigned to $(q, d)$ by the DE model.

### 3.2 SCORE-BASED DISTILLATION FOR IR MODELS

Most distillation schemes for IR (e.g., Chen et al., 2021; Lu et al., 2020; Hofstätter et al., 2020; Ren et al., 2021; Santhanam et al., 2021) rely on teacher relevance scores (cf. Fig. 1). In particular, given a training set $\mathcal{S}_n$ and a teacher with scorer $s^{\mathrm{t}}$, one learns a student with scorer $s^{\mathrm{s}}$ by minimizing

$$R(s^{\mathrm{s}}, s^{\mathrm{t}}; \mathcal{S}_n) = \frac{1}{n} \sum\nolimits_{i \in [n]} \ell_{\mathrm{d}}(s^{\mathrm{s}}_{q, \mathbf{d}_i}, s^{\mathrm{t}}_{q, \mathbf{d}_i}), \tag{3}$$

where $\ell_{\mathrm{d}}$ captures the discrepancy between $s^{\mathrm{s}}$ and $s^{\mathrm{t}}$. See Appendix A for common choices for $\ell_{\mathrm{d}}$.

## 4 EMBEDDING-MATCHING BASED DISTILLATION

Since modern neural IR models are inherently embedding-based, we propose to explicitly align the embedding spaces of the teacher and student via a novel distillation method, namely EmbedDistill. Note that our proposal goes beyond existing distillation methods in the IR literature that only utilize the teacher scores. Next, we describe EmbedDistill for two prevalent settings: (1) distilling a large DE model to a smaller DE model;[2] and (2) distilling a CE model to a DE model.

### 4.1 DE TO DE DISTILLATION

Given a $(q, d)$ pair, let $\texttt{emb}_q^{\mathrm{t}}$ and $\texttt{emb}_d^{\mathrm{t}}$ be the query and document embeddings produced by the query encoder $\mathrm{Enc}_Q^{\mathrm{t}}$ and document encoder $\mathrm{Enc}_D^{\mathrm{t}}$ of the teacher DE model, respectively. Similarly, let

---

[1] It is common to employ dual-encoder models where query and document encoders are shared.

[2] We focus on DE to DE distillation setup as the CE to CE distillation is special case of the former with the classification vector $w$ (cf. Eq. 2) being the trivial second encoder.

$\mathsf{emb}_q^{\mathsf{s}}$ and $\mathsf{emb}_d^{\mathsf{s}}$ denote the query and document embeddings produced by a student DE model with $(\mathrm{Enc}_Q^{\mathsf{s}}, \mathrm{Enc}_D^{\mathsf{s}})$ as its query and document encoders. Now, EmbedDistill optimizes the following embedding alignment loss in addition to the score-matching loss from Sec. 3.2:

$$R_{\mathrm{Emb}}(\mathsf{t}, \mathsf{s}; \mathcal{S}_n) = \frac{1}{n} \sum\nolimits_{(q,d) \in \mathcal{S}_n} \left( \|\mathsf{emb}_q^{\mathsf{t}} - \mathrm{proj}(\mathsf{emb}_q^{\mathsf{s}})\| + \|\mathsf{emb}_d^{\mathsf{t}} - \mathrm{proj}(\mathsf{emb}_d^{\mathsf{s}})\| \right), \quad (4)$$

where $\mathrm{proj}$ is an optional trainable layer that is required if the teacher and student produce different sized embeddings. Alternatively, one can employ other variants of EmbedDistill, e.g., focusing on only aligning the query embeddings takes the following form (cf. Fig. 2).

$$R_{\mathrm{Emb},Q}(\mathsf{t}, \mathsf{s}; \mathcal{S}_n) = \frac{1}{n} \sum\nolimits_{q \in \mathcal{S}_n} \|\mathsf{emb}_q^{\mathsf{t}} - \mathrm{proj}(\mathsf{emb}_q^{\mathsf{s}})\|. \quad (5)$$

**Asymmetric DE.** We also propose a novel student DE configuration where the student employs the teacher's document encoder (i.e., $\mathrm{Enc}_D^{\mathsf{s}} = \mathrm{Enc}_D^{\mathsf{t}}$) and only train its query encoder, which is much smaller compared to the teacher's query encoder. For such a setting, it is natural to only employ the embedding matching loss in Eq. 5 as the document embeddings are aligned by design (cf. Fig. 2).

Note that this asymmetric student DE does not incur an increase in latency despite use of large teacher document encoder. This is because the large document encoder is only needed to create a good quality document index offline, and only the query encoder is evaluated at inference time. Thus, for DE to DE distillation, we prescribe the asymmetric DE configuration universally. Our theoretical analysis and experimental results suggest that the ability to inherit the document tower from the teacher DE model can drastically improve the final performance, especially when combined with EmbedDistill.

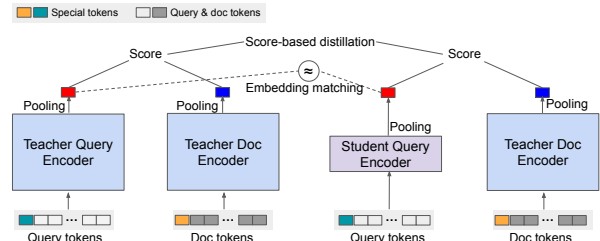

Figure 2: Proposed DE to DE distillation with query embedding matching. The figure describes a setting where student employs an asymmetric DE configuration with a small query encoder and a large (non-trainable) document encoder inherited from the teacher. The smaller query encoder ensures small latency for encoding query during inference, and large document encoder leads to a good quality document index.

## 4.2 CE TO DE DISTILLATION

Let $\mathrm{Enc}^t$ denote the single teacher CE model encoder, and $(\mathrm{Enc}_Q^{\mathsf{s}}, \mathrm{Enc}_D^{\mathsf{s}})$ denote the student DE model's query and document encoders. When distilling from a CE to DE model, defining an effective embedding matching task is not as straightforward as in Sec. 4.1: since CE models jointly encode query-document pairs, it is not obvious how to extract individual query and document embeddings.

As a naïve solution, for a given $(q, d)$ pair, one can simply match a joint transformation of the student's query embedding $\mathsf{emb}_q^{\mathsf{s}}$ and document embedding $\mathsf{emb}_d^{\mathsf{s}}$ to the teacher's joint embeddings $\mathsf{emb}_{q,d}^{\mathsf{t}}$. However, we observed that including such an embedding matching task often leads to severe over-fitting, and results in a poorly generalizable student. Since $s^{\mathsf{t}}(q, d) = \langle w, \mathsf{emb}_{q,d}^{\mathsf{t}} \rangle$, during CE model training, the joint embeddings $\mathsf{emb}_{q,d}^{\mathsf{t}}$ for relevant and irrelevant $(q, d)$ pairs are encourage to be aligned with $w$ and $-w$, respectively. This produces degenerate embeddings that do not capture semantic query-to-query or document-to-document relationships. We notice that even the final query and document token embeddings lose such semantic structure. Thus, a teacher CE model with $s^{\mathsf{t}}(q, d) = \langle w, \mathsf{emb}_{q,d}^{\mathsf{t}} \rangle$ does not add value for distillation beyond score-matching; in fact, it *hurts* to include naïve embedding matching. Next, we propose a modified CE model training strategy that facilitates EmbedDistill.

**CE models with dual pooling.** We propose *dual pooling* to produce two embeddings $\mathsf{emb}_{q \leftarrow (q,d)}^{\mathsf{t}}$ and $\mathsf{emb}_{d \leftarrow (q,d)}^{\mathsf{t}}$ from a CE model that serve as the *proxy* query and document embeddings, respectively. Accordingly, we define the relevance score as $s^{\mathsf{t}}(q, d) = \langle \mathsf{emb}_{q \leftarrow (q,d)}^{\mathsf{t}}, \mathsf{emb}_{d \leftarrow (q,d)}^{\mathsf{t}} \rangle$. We explore two variants of dual pooling: (1) special token-based pooling that pools from [CLS] and [SEP]; and (2) segment-based weighted mean pooling that separately performs weighted averaging on the query and document segments of the final token embeddings. See Appendix B for details.

In addition to employing the dual pooling, we also utilize a reconstruction loss during the CE training, which measures the likelihood of predicting each token of the original input from the final token embeddings. This loss encourages reconstruction of query and document tokens based on the final token embeddings and prevents the degeneration of the token embeddings during training on the IR task. Given proxy embeddings from the teacher CE , we can perform EmbedDistill with the loss defined in Eq. 4 or Eq. 5 (cf. Fig. 3).

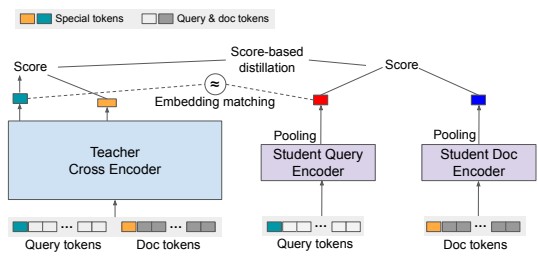

Figure 3: Illustration of CE to DE distillation using EmbedDistill, with CE model employing dual pooling.

### 4.3 TASK-SPECIFIC ONLINE DATA GENERATION

Data augmentation as a general technique has been previously considered in the IR literature (see, e.g., Nogueira et al., 2019b; Oğuz et al., 2021; Izacard et al., 2021; Ma et al., 2021), especially in data-limited, out-of-domain, or zero-shot settings. Since EmbedDistill aims to align the embeddings spaces of the teacher and student, the ability to generate similar queries or documents can naturally help enforce such an alignment globally on the task-specific manifold. Given a set of unlabeled task-specific query and document pairs $\mathcal{U}_m$, we can further add the embedding-alignment loss $R_{\mathrm{Emb}}(t, s; \mathcal{U}_m)$ to our training objective (cf. Eq.4). Interestingly, for DE to DE distillation setting, our approach can even benefit from a large collection of task-specific queries $\mathcal{Q}'$ or documents $\mathcal{D}'$. Here, as we can independently add the additional embedding matching losses $R_{\mathrm{Emb,Q}}(t, s; \mathcal{Q}')$ and $R_{\mathrm{Emb,D}}(t, s; \mathcal{D}')$ that focus on queries and documents, respectively.

## 5 IMPROVEMENTS IN THE GENERALIZATION OF STUDENT

Note that we motivate EmbedDistill as well as asymmetric DE configuration where the student DE model inherits the teacher's document encoder from their potential ability to ensure a better alignment between the teacher and student embedding spaces. In this section, we provide a theoretical justification for both of these proposals by showing that they indeed result in a better generalization performance for the student and reduce the gap between the teacher and the student. We break our analysis into two steps: 1) First, we show that, using EmbedDistill and inheriting the document encoder from the teacher, the student's empirical risk (as measured by the distillation objective) gets closer to the teacher's population risk; and 2) Second, we argue that both of these techniques have favorable implications on the distillation loss of the student via uniform deviation bounds.

The following result studies the gap between student's expected empirical (distillation) risk and teacher's population risk (see Appendix C.1 for a formal statement and proof). For simplicity, we focus on $L = 1$ (cf. Sec. 3). The result can be extended to $L > 1$ with more complex notation.

**Proposition 1** (Expected risk bound (informal)). *Let label $y \in \{0, 1\}$ indicate the relevance of query-document pair $(q, d)$. Suppose that the score-based distillation loss $\ell_{\mathrm{d}}$ in Eq. 3 is based on binary cross entropy loss (Eq. 11 in Appendix A). Let one-hot (label-dependent) loss $\ell$ in Eq. 1 be the binary cross entropy loss (Eq. 9 in Appendix A). Assume that all encoders have the same output dimension. Under regularity conditions, we have*

$$\mathbb{E}\left[R(s^{\mathrm{s}}, s^{\mathrm{t}}; \mathcal{S}_n) - \mathbb{E}\ell(s^{\mathrm{t}}_{q,d}, y)\right] = \mathcal{O}\Big(\mathbb{E}_d[\|\mathsf{emb}^{\mathrm{s}}_d - \mathsf{emb}^{\mathrm{t}}_d\|_2] + \mathbb{E}_q[\|\mathsf{emb}^{\mathrm{s}}_q - \mathsf{emb}^{\mathrm{t}}_q\|_2]$$
$$+ \mathbb{E}_{(q,d,y)}\big|\mathrm{sigmoid}(\langle \mathsf{emb}^{\mathrm{t}}_q, \mathsf{emb}^{\mathrm{t}}_d \rangle) - y\big|\Big).$$

Proposition 1 can be viewed as the *bias* of the student wrt. the teacher, as realized by the distillation. It shows that the bias can be upper bounded by three terms:. 1) the expected difference between the doc embeddings of the student and the teacher, 2) the expected difference between the query embeddings, and 3) the teacher's error in modeling the true label probability. Observe that the student's bias is smaller when the embeddings of the student match those of the teacher. In particular, when the student inherits the document encoder from the teacher (as in Fig. 2), the error in the first

| Method | Recall@5 | | Recall@20 | | Recall@100 | |
|---|---|---|---|---|---|---|
| | **82M** | **16M** | **82M** | **16M** | **82M** | **16M** |
| Train student directly | 41.9 | 39.5 | 64.5 | 59.9 | 82.0 | 76.3 |
| + Distill from teacher | 48.3 | 44.9 | 67.2 | 61.1 | 80.9 | 74.8 |
| + Inherit document embeddings | 56.9 | 47.2 | 74.3 | 64.0 | 85.4 | 77.0 |
| + Query embedding matching | 61.8 | 56.7 | 78.7 | 74.6 | 89.0 | 85.9 |
| + Query generation | 61.7 | 57.1 | 79.4 | 75.2 | 89.6 | 86.7 |
| Train student only using embedding matching and inherit doc embeddings | 63.7 | 57.9 | 80.3 | 74.6 | 90.3 | 85.7 |
| + Query generation | 64.1 | 58.9 | 80.5 | 76.0 | 90.4 | 86.6 |

Table 1: *Full* recall performance of various student DE models on NQ dev set, including symmetric DE student model (82M or 16M transformer for both encoders), and asymmetric DE student model (82M or 16M transformer as query encoder and document embeddings inherited from the teacher). All distilled students used the same teacher (110M parameter RoBERTa-base models as both encoders), with the full Recall@5 = 64.6, Recall@20 = 81.7, and Recall@100 = 91.5.

term vanishes. These observations also justify EmbedDistill which either employ Eq. 4 or Eq. 5 (when student inherits teacher's document encoder).

Now, we analyze the deviation of a student model from its training empirical risk at the test time, which is typically captured by the uniform deviation bounds based on quantities like Rademacher complexity. Again, we restrict ourselves to binary cross-entropy loss with $L = 1$ for simplicity. Due to space constraints, we present an informal statement of the result (see Appendix C.2 for a more precise statement and proof).

**Proposition 2.** *Let $\ell_d$ be a distillation loss which is $L_{\ell_d}$-Lipschitz in its first argument. Let $\mathcal{F}$ and $\mathcal{G}$ denote the function classes for the query and document encoders, respectively. Further assume that, for each query and document encoder in our function class, the query and document embeddings have their $\ell_2$-norm bounded by $K$. Then, we have the following uniform deviation bounds*

$$\sup_{s^s \in \mathcal{F} \times \mathcal{G}} \frac{1}{n} \sum_{i \in [n]} \ell_d\big(s^s_{q_i,d_i}, s^t_{q_i,d_i}\big) - \mathbb{E}\ell_d\big(s^s_{q,d}, s^t_{q,d}\big) \leq \mathbb{E}_{\mathcal{S}_n} \frac{48 K L_{\ell_d}}{\sqrt{n}} \int_0^\infty \sqrt{\log\big(N(u, \mathcal{F})N(u, \mathcal{G})\big)}\, du; \quad (6)$$

$$\sup_{s^s \in \mathcal{F} \times \{g^*\}} \frac{1}{n} \sum_{i \in [n]} \ell_d\big(s^s_{q_i,d_i}, s^t_{q_i,d_i}\big) - \mathbb{E}\ell_d\big(s^s_{q,d}, s^t_{q_i,d_i}\big) \leq \mathbb{E}_{\mathcal{S}_n} \frac{48 K L_{\ell_d}}{\sqrt{n}} \int_0^\infty \sqrt{\log N(u, \mathcal{F})}\, du. \quad (7)$$

*Here, $g^*$ is a fixed document encoder and $N(u, \cdot)$ denotes the $u$-covering number of a function class.*

Note that Eq. 6 and Eq. 7 corresponds to uniform deviation when we train *without* and *with* a frozen document encoder, respectively. It is clear that the bound in Eq. 7 is lower than Eq. 6 (due to absence of $N(u, \mathcal{G})$ term which is always larger than 1), which alludes to desirable impact of employing a frozen document encoder (in terms of deviation in train and test performance). When we further employ EmbedDistill (e.g., with the loss in Eq. 5), it regularizes the function class of query encoders; effectively, reducing it to $\mathcal{F}'$ with $|\mathcal{F}'| \leq |\mathcal{F}|$. This has further desirable implication for reducing the uniform deviation bounds as $N(u, \mathcal{F}') \leq N(u, \mathcal{F})$.

# 6 EXPERIMENTS

We now conduct a comprehensive evaluation of the proposed distillation approach. Specifically, we highlight the utility of the proposed approach for both DE to DE and CE to DE distillation. We also showcase the benefits of combining our distillation approach with query generation methods.

## 6.1 EXPERIMENTAL SETUP

**Benchmarks and evaluation metrics.** We focus on two popular IR benchmarks — Natural Questions (NQ; Kwiatkowski et al. 2019b) and MSMARCO (Nguyen et al., 2016), which focus on finding the most relevant passage/document given a question and a search query, respectively. NQ provides both standard test and dev sets, whereas MSMARCO has only standard dev set publicly available. In what follows, we use the terms query (document) and question (passages) interchangeably. For NQ, we use the standard full recall (*strict*) as well as the *relaxed* recall metric (Karpukhin et al., 2020) to evaluate the retrieval performance. For MSMARCO, we focus on the standard metrics *Mean Reciprocal Rank* (MRR)@10, and *normalized Discounted Cumulative Gain* (nDCG)@10. See Appendix D for a detailed discussion on these evaluation metrics.

**Model architectures.** We follow the standard Transformers-based IR model architectures similar to Karpukhin et al. (2020); Qu et al. (2021); Oğuz et al. (2021). Our CE model is based on a RoBERTa-base model (Liu et al. (2019); 12-layer, 768 dim, 124M parameters). We utilized various sizes of DE models based on RoBERTa-base, DistilRoBERTa (Sanh et al. (2019); 6-layer, 768 dim, 82M parameters – 2/3 of base), or RoBERTa-mini (Turc et al. (2019); 4-layer, 256 dim, 16M parameters – 1/8 of base). For query generation (cf. Sec. 4.3), we employ BART-base (Lewis et al., 2020), an encoder-decoder model, to generate similar questions from each training

| Method | MRR@10 | | nDCG@10 | |
|---|---|---|---|---|
| | **82M** | **16M** | **82M** | **16M** |
| Train student directly | 29.7 | 26.3 | 35.2 | 31.4 |
|   + Distill from teacher | 31.6 | 28.4 | 37.2 | 33.5 |
|   + Inherit doc embeddings | 32.4 | 30.2 | 38.0 | 35.8 |
|   + Query embedding matching | 32.8 | 31.9 | 38.6 | 37.6 |
|   + Query generation | 33.0 | 32.0 | 38.8 | 37.7 |
| Train student only using embedding matching and inherit doc embeddings | 32.7 | 31.8 | 38.5 | 37.5 |
|   + Query generation | 33.0 | 31.8 | 38.9 | 37.5 |

Table 2: Performance of various DE models on MSMARCO dev set. A teacher model (110M parameter RoBERTa-base models as both encoders) achieving MRR@10 of 33.1 and nDCG@10 of 38.8 is used. The table shows performance of the symmetric DE student model (82M or 16M transformer as both encoders), and asymmetric DE student model (82M or 16M transformer as query encoder and document embeddings inherited from the teacher).

example's input question (query). We randomly mask $10\%$ of tokens and inject zero mean Gaussian noise with $\sigma = \{0.1, 0.2\}$ between the encoder and decoder. See Appendix E for details.

## 6.2 DE TO DE DISTILLATION

For both NQ and MSMARCO, teacher DE models are based on RoBERTa-base model (see Appendix F for the training details). For DE to DE distillation, we consider two kinds of configurations for the student DE model: (1) *Symmetric*. We use identical question and document encoders. We evaluate DistilRoBERTa and RoBERTa-mini for both datasets. (2) *Asymmetric*. The student DE model inherits its document embeddings from the teacher DE model, which *are not* trained during the distillation. For query encoder, we use DistilRoBERTa or RoBERTa-mini which are smaller than the teacher document encoder.

**Student DE model training.** We train student DE models using a combination of (i) one-hot loss (cf. Eq. 8 in Appendix A) on training data; (ii) distillation loss in (cf. Eq. 10 in Appendix A); and (iii) embedding matching loss in Eq. 5. We used [CLS]-pooling for all student encoders. Unlike DPR (Karpukhin et al., 2020), we do not use hard negatives from BM25 or other models, which greatly simplifies our distillation procedure.

**Results and discussion.** To understand the impact of various proposed configurations and losses, we train models by sequentially adding components and evaluate on NQ and MSMARCO dev set as shown in Table 1 and 2 , respectively. (See Table 5 in Appendix G.1 for performance on NQ in terms of the relaxed recall.) We begin by training a symmetric DE without distillation. As expected, moving to distillation brings in considerable gains.

Next, we swap the student document encoder with document embeddings from the teacher (non-trainable), which leads to a good jump in the performance. Now we can introduce EmbedDistill with Eq. 5 for aligning query representations between student and teacher. The two losses are combined with weight of $1.0$ and $5.0$ for NQ and MSMARCO, respectively. This improves performance significantly, e.g., it provides $\sim 5$ and $\sim 9$ points increase in recall@5 on NQ with students based on DistilRoBERTa and RoBERTa-mini, respectively. We further explore the utility of EmbedDistill in aligning the teacher and student embedding spaces in Appendix H.1.

On top of the two losses (standard distillation and embedding matching), we also employ $R_{\text{Emb},Q}(t, s; Q')$ from Sec. 4.3 on 2 additional questions (per input question) generated from BART for further gain. We also try a variant where we eliminate the standard distillation loss and only employ the embedding matching loss in Eq. 5 along with inheriting document embedding from the teacher. This configuration without the standard distillation loss leads to excellent performance (with query generation again providing additional gains).

It is worth highlighting that DE models trained with the proposed methods (e.g. asymmetric DE with embedding matching and generation) achieve 99% of the performance in both NQ/MSMARCO tasks with a query encoder that is half the size of that of the teacher. Furthermore, even with 1/8th

size of the query encoder, our proposal can achieve 91-97% of the performance. This is particularly useful for latency critical applications with minimal impact on the final performance.

Finally, we take our best student models for NQ based on the dev set, i.e., one trained using only embedding matching loss with inherited document embedding and using data augmentation from query generation, and evaluate it on the NQ test set (cf. Table 3). We compare with various prior work and note they worked with considerably bigger models in terms of depth (12 or 24 layers) and width (upto 1024 dims), while our approach obtains competitive performance with fewer than 50% of the parameters.

| Method | #Layers | R@20 | R@100 |
|---|---|---|---|
| DPR (Karpukhin et al., 2020) | 12 | 78.4 | 85.4 |
| DPR + PAQ (Oğuz et al., 2021) | 12 | 84.0 | 89.2 |
| DPR + PAQ (Oğuz et al., 2021) | 24 | 84.7 | 89.2 |
| ACNE (Xiong et al., 2021) | 12 | 81.9 | 87.5 |
| RocketQA (Qu et al., 2021) | 12 | 82.7 | 88.5 |
| MSS-DPR (Sachan et al., 2021) | 12 | 84.0 | 89.2 |
| MSS-DPR (Sachan et al., 2021) | 24 | 84.8 | 89.8 |
| Our teacher | 12 (2×124M) | 80.7 | 87.3 |
| Student w/ proposed method | 6 (82M) | 80.4 | 86.8 |
| Student w/ proposed method | 4 (16M) | 77.4 | 85.3 |

Table 3: Performance of EmbedDistill for DE to DE distillation on NQ test set. Note that the prior work mentioned in the table rely on techniques such as negative mining and multi-stage training. In contrast, we explore the orthogonal direction of embedding-matching that improves *single-stage* distillation, which can be combined with the aforementioned techniques.

Note that, even with 6 layers, our student model is very close (99%) to its teacher.

## 6.3 CE TO DE DISTILLATION

We consider two CE teachers for MSMARCO: a standard [CLS]-pooled CE teacher, and the Dual-pooled CE teacher (cf. Sec. 4.2). Both teachers are based on RoBERTa-base and trained on triples in the training set for 300K steps with cross-entropy loss.

**Student DE model training.** We considered the following distillation variants: standard score-based distillation from the [CLS]-pooled teacher, and our novel Dual-pooled CE teacher (with and without embedding matching

| Method | MRR@10 |
|---|---|
| [CLS]-pooled teacher | 37.1 |
| Dual-pooled teacher | 37.0 |
| Standard distillation from [CLS]-pooled teacher | 33.0 |
| +Joint matching | 32.4 |
| Standard distillation from Dual-pooled teacher | 33.3 |
| +Query matching | 33.7 |

Table 4: Performance of DE models distilled from [CLS]-pooled and Dual-pooled CE models on MSMARCO. While both teacher models perform similarly, embedding matching-based distillation only works with the Dual-pooled teacher. See Appendix G for nDCG@10 metric.

loss). For each variant, we initialize encoders of the student DE model with two RoBERTa-base models and train for 500K steps on the training triples. We performed the naïve joint embedding matching for the [CLS]-pooled teacher (cf. Sec. 4.2) and employed the query embedding matching (cf. Eq.5) for the Dual-pooled CE teacher. In either case, embedding-matching loss is added on top of the standard cross entropy loss with the weight of 1.0 (when used).

**Results and discussion.** Table 4 evaluates the effectiveness of the dual pooling and the embedding matching for CE to DE distillation. As described in Sec. 4.2, the traditional [CLS]-pooled teacher did not provide any useful embedding for the embedding matching (see Appendix H.2 for the further analysis of the resulting embedding space). However, with the Dual-pooled teacher, embedding matching does boost student's performance.

## 7 CONCLUSION

We propose EmbedDistill — a novel distillation method for IR that goes beyond simple score matching. We specialize it to distill a DE model into another DE model by (a) reusing the teacher's document encoder in the student and (b) aligning query embeddings of the teacher and student. This simple approach delivers immediate quality and computational gains in practical deployments and we demonstrate them on MSMARCO and NQ benchmarks. We show that query generation technique further improves the performance of the distilled student. We generalize the proposed approach to distill a CE model to a DE model and show the benefits on MSMARCO. Finally, our theoretical analysis alludes to the favorable implications of both embedding matching and inheriting document encoder in DE to DE distillation setting. A more comprehensive and systematic analysis of embedding matching-based distillation for IR is an exciting avenue for future research.

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

## A   LOSS FUNCTIONS

Here, we state various (per-example) loss functions that most commonly define training objectives for IR models. Typically, one hot training with original label is performed using *softmax-based cross-entropy loss* functions:

$$\ell\big(s_{q,\mathbf{d}_i}, \mathbf{y}_i\big) \;=\; -\sum_{j\in[L]} y_{i,j} \cdot \log\Big(\frac{\exp(s(q_i, d_{i,j}))}{\sum\limits_{j'\in[L]} \exp(s(q_i, d_{i,j'}))}\Big). \tag{8}$$

Alternatively, it's also common to employ an one-vs-all loss function based on *binary cross-entropy loss* as follows:

$$\ell\big(s_{q,\mathbf{d}_i}, \mathbf{y}_i\big) \;=\; -\sum_{j\in[L]} \bigg( y_{i,j} \cdot \log\Big(\frac{1}{1\,+\,\exp(-s(q_i, d_{i,j}))}\Big) \;+$$
$$(1 - y_{i,j}) \cdot \log\Big(\frac{1}{1\,+\,\exp(s(q_i, d_{i,j}))}\Big)\bigg). \tag{9}$$

As for distillation, one can define a distillation objective based on the softmax-based cross-entropy loss as[3]:

$$\ell_{\mathrm{d}}\big(s^{\mathrm{s}}_{q,\mathbf{d}_i}, s^{\mathrm{t}}_{q,\mathbf{d}_i}\big) = -\sum_{j\in[L]} \bigg( \frac{\exp(s^{\mathrm{t}}_{i,j})}{\sum_{j'\in[L]} \exp(s^{\mathrm{t}}_{i,j'})} \cdot \log\Big(\frac{\exp(s^{\mathrm{s}}_{i,j})}{\sum_{j'\in[L]} \exp(s^{\mathrm{s}}_{i,j'})}\Big)\bigg), \tag{10}$$

where $s^{\mathrm{t}}_{i,j} := s^{\mathrm{t}}(q_i, d_{i,j})$ and $s^{\mathrm{s}}_{i,j} := s^{\mathrm{s}}(q_i, d_{i,j})$ denote the teacher and student scores, respectively. On the other hand, the distillation objective with the binary cross-entropy takes the form:

$$\ell_{\mathrm{d}}\big(s^{\mathrm{s}}_{q,\mathbf{d}_i}, s^{\mathrm{t}}_{q,\mathbf{d}_i}\big) \;=\; -\sum_{j\in[L]} \bigg( \frac{1}{1\,+\,\exp(-s^{\mathrm{t}}_{i,j})} \cdot \log\Big(\frac{1}{1\,+\,\exp(-s^{\mathrm{s}}_{i,j})}\Big) \;+$$
$$\frac{1}{1\,+\,\exp(s^{\mathrm{t}}_{i,j})} \cdot \log\Big(\frac{1}{1\,+\,\exp(s^{\mathrm{s}}_{i,j})}\Big)\bigg). \tag{11}$$

Finally, distillation based on the meas square error (MSE) loss (aka. logit matching) employs the following loss function:

$$\ell_{\mathrm{d}}\big(s^{\mathrm{s}}_{q,\mathbf{d}_i}, s^{\mathrm{t}}_{q,\mathbf{d}_i}\big) = \sum_{j\in[L]} \big(s^{\mathrm{t}}(q_i, d_{i,j}) - s^{\mathrm{s}}(q_i, d_{i,j})\big)^2. \tag{12}$$

## B   DUAL POOLING DETAILS

In this work, we focus on two kinds of dual pooling strategies:

- **Special tokens-based dual pooling.** Let $\mathrm{pool}_{\mathrm{CLS}}$ and $\mathrm{pool}_{\mathrm{SEP}}$ denote the pooling operations that return the embeddings of the `[CLS]` and `[SEP]` tokens, respectively. We define
$$\mathrm{emb}^{\mathrm{t}}_{q\leftarrow(q,d)} = \mathrm{pool}_{\mathrm{CLS}}\big(\mathrm{Enc}^{\mathrm{t}}(\tilde{o})\big),$$
$$\mathrm{emb}^{\mathrm{t}}_{d\leftarrow(q,d)} = \mathrm{pool}_{\mathrm{SEP}}\big(\mathrm{Enc}^{\mathrm{t}}(\tilde{o})\big), \tag{13}$$
where $\tilde{o}$ denotes the input token sequence to the Transformers-based encoder, which consists of { query, document, special } tokens.

- **Segment-based weighted-mean dual pooling.** Let $\mathrm{Enc}^{\mathrm{t}}(\tilde{o})|_Q$ and $\mathrm{Enc}^{\mathrm{t}}(\tilde{o})|_D$ denote the final query token embeddings and document token embeddings produced by the encoder, respectively. We define the *proxy* query and document embeddings
$$\mathrm{emb}^{\mathrm{t}}_{q\leftarrow(q,d)} = \mathrm{mean}_{\mathrm{wt}}\big(\mathrm{Enc}^{\mathrm{t}}(\tilde{o})|_Q\big),$$
$$\mathrm{emb}^{\mathrm{t}}_{d\leftarrow(q,d)} = \mathrm{mean}_{\mathrm{wt}}\big(\mathrm{Enc}^{\mathrm{t}}(\tilde{o})|_D\big), \tag{14}$$
where $\mathrm{mean}_{\mathrm{wt}}(\cdot)$ denotes the weighted mean operation. We employ the specific weighting scheme where each token receives a weight equal to the inverse of the square root of the token-sequence length.

---

[3]It is common to employ temperature scaling with softmax operation. We do not explicitly show the temperature parameter for ease of exposition.

## C   Deferred details and poofs from Section 5

In this section we present more precise statements and proofs of Proposition 1 and 2 (stated informally in Section 5 of the main text) along with the necessary background. First, for the ease of exposition, we define new notation which will facilitate theoretical analysis in this section.

**Notation**   Denote the query and document encoders as $f\colon \mathcal{Q} \to \mathbb{R}^k$ and $g\colon \mathcal{D} \to \mathbb{R}^k$ for the student, and $F\colon \mathcal{Q} \to \mathbb{R}^k, G\colon \mathcal{D} \to \mathbb{R}^k$ for the teacher (in the dual-encoder setting). With $q$ denoting a query and $d$ denoting a document, $f(q)$ and $g(d)$ then denote query and document embeddings by the student. We define $F(q)$ and $G(d)$ similarly for embeddings by the teacher.[4]

### C.1   Bound on the expected risk

**Proposition 3** (Formal statement of Proposition 1). *Given an example $(q, d, y) \in \mathcal{Q} \times \mathcal{D} \times \{0, 1\}$, let $s^{f,g}(q, d) = f(q)^T g(d)$ be the scores assigned to the $(q, d)$ pair by a dual-encoder model with $f$ and $g$ as query and document encoders, respectively. Let $\ell$ and $\ell_{\mathrm{d}}$ be the binary cross-entropy loss (cf. Eq. 9 with $L = 1$) and the distillation-specific loss based on it (cf. Eq. 11 with $L = 1$), respectively. In particular,*

$$\ell(s^{F,G}(q,d), y) := -y \log \sigma\left(F(q)^\top G(d)\right) - (1-y) \log\left[1 - \sigma\left(F(q)^\top G(d)\right)\right]$$

$$\ell_{\mathrm{d}}(s^{f,g}(q,d), s^{F,G}(q,d)) := -\sigma\left(F(q)^\top G(d)\right) \cdot \log \sigma\left(f(q)^\top g(d)\right) - $$
$$\left[1 - \sigma\left(F(q)^\top G(d)\right)\right] \cdot \log\left[1 - \sigma\left(f(q)^\top g(d)\right)\right],$$

*where $\sigma$ is the sigmoid function. Assume that*

1. *All encoders $f, g, F$, and $G$ have the same output dimension $k \geq 1$.*

2. *There exist $K_Q, K_D \in (0, \infty)$ such that $\sup_{q \in \mathcal{Q}} \max\left\{\|f(q)\|_2, \|F(q)\|_2\right\} \leq K_Q$ and $\sup_{d \in \mathcal{D}} \max\left\{\|g(d)\|_2, \|G(d)\|_2\right\} \leq K_D$.*

*Given a sample $\{(q_i, d_i, y_i)\} \stackrel{i.i.d.}{\sim} \mathbb{P}(q, d, y)$, we have*

$$\mathbb{E}\left[\frac{1}{n}\sum_{i=1}^n \ell_{\mathrm{d}}\left(s^{f,g}(q_i, d_i), s^{F,G}(q_i, d_i)\right) - \mathbb{E}\ell\left(s^{F,G}(q, d), y\right)\right] \leq 2K_Q \mathbb{E}\left[\|g(d) - G(d)\|_2\right] + $$
$$2K_D \mathbb{E}\left[\|f(q) - F(q)\|_2\right] + K_Q K_D \mathbb{E}_{(q,d,y)}\left|\sigma\left(F(q)^\top G(d)\right) - y\right|.$$

*Proof of Proposition 3.* We first note that the distillation loss can be rewritten as

$$\ell_{\mathrm{d}}\left(s^{f,g}(q,d), s^{F,G}(q,d)\right) = \left(1 - \sigma(F(q)^\top G(d)\right) f(q)^\top g(d) + \gamma(-f(q)^\top g(d)),$$

where $\gamma(v) := \log[1 + e^v]$ is the softplus function. Similarly, the one-hot (label-dependent) loss can be rewritten as

$$\ell\left(s^{F,G}(q,d), y\right) = (1-y)F(q)^\top G(d) + \gamma(-F(q)^\top G(d)).$$

As a shorthand, define

$$\tilde{R}(f, g) := \frac{1}{n}\sum_{i=1}^n \ell_{\mathrm{d}}\left(s^{f,g}(q_i, d_i), s^{F,G}(q_i, d_i)\right),$$

$$R(F, G) := \mathbb{E}_{(q,d,y)\sim\mathbb{P}(q,d,y)}\left[\ell\left(s^{F,G}(q,d), y\right)\right],$$

as the empirical risk based on the distillation loss, and the population risk based on the label-dependent loss, respectively. With this notation, the quantity to upper bound can be rewritten as

$$\mathbb{E}\left[\tilde{R}(f, g) - R(F, G)\right] = \mathbb{E}\left[\overbrace{\tilde{R}(f, g) - \tilde{R}(f, G)}^{:=\square_1} + \overbrace{\tilde{R}(f, G) - \tilde{R}(F, G)}^{:=\square_2} + \overbrace{\tilde{R}(F, G) - R(F, G)}^{:=\square_3}\right].$$
$$(15)$$

---

[4]Note that, as per the notations in the main text, we have $(f, g) = (\mathrm{Enc}_Q^{\mathrm{s}}, \mathrm{Enc}_D^{\mathrm{s}})$ and $(F, G) = (\mathrm{Enc}_Q^{\mathrm{t}}, \mathrm{Enc}_D^{\mathrm{t}})$. Similarly, we have $(\mathrm{emb}_q^{\mathrm{t}}, \mathrm{emb}_d^{\mathrm{t}}) = (f(q), g(d))$ and $(\mathrm{emb}_q^{\mathrm{t}}, \mathrm{emb}_d^{\mathrm{t}}) = (F(q), G(d))$.

We start by bounding $\mathbb{E}[\square_1]$ as

$$
\begin{aligned}
\mathbb{E}[\square_1] &= \mathbb{E}\left[\frac{1}{n}\sum_{i=1}^{n}\ell_{\mathrm{d}}\big(s^{f,g}(q_i,d_i),s^{F,G}(q_i,d_i)\big) - \frac{1}{n}\sum_{i=1}^{n}\ell_{\mathrm{d}}\big(s^{f,G}(q_i,d_i),s^{F,G}(q_i,d_i)\big)\right] \\
&= \mathbb{E}\left[\ell_{\mathrm{d}}\big(s^{f,g}(q,d),s^{F,G}(q,d)\big) - \ell_{\mathrm{d}}\big(s^{f,G}(q,d),s^{F,G}(q,d)\big)\right] \\
&= \mathbb{E}\left[\big(1-\sigma(F(q)^\top G(d))\big)\,f(q)^\top g(d) + \gamma(-f(q)^\top g(d))\right] \\
&\quad - \mathbb{E}\left[\big(1-\sigma(F(q)^\top G(d))\big)\,f(q)^\top G(d) + \gamma(-f(q)^\top G(d))\right] \\
&= \mathbb{E}\left[f(q)^\top(g(d)-G(d))\big(1-\sigma(F(q)^\top G(d))\big) + \gamma(-f(q)^\top g(d)) - \gamma(-f(q)^\top G(d))\right] \\
&\overset{(a)}{\leq} \mathbb{E}\left[f(q)^\top(g(d)-G(d))\big(1-\sigma(F(q)^\top G(d))\big) + \big|f(q)^\top g(d) - f(q)^\top G(d)\big|\right] \\
&\overset{(b)}{\leq} \mathbb{E}\left[\|f(q)\|\|g(d)-G(d)\|\big(1-\sigma(F(q)^\top G(d))\big) + \|f(q)\|\|g(d)-G(d)\|\right] \\
&\leq K_Q\mathbb{E}\left[\|g(d)-G(d)\|_2\big(2-\sigma(F(q)^\top G(d))\big)\right] \\
&\leq 2K_Q\mathbb{E}\left[\|g(d)-G(d)\|_2\right]
\end{aligned}
\tag{16}
$$

where at $(a)$ we use the fact that $\gamma$ is a Lipschitz continuous function with Lipschitz constant 1, and at $(b)$ we use Cauchy-Schwarz inequality.

Similarly for $\mathbb{E}[\square_2]$, we proceed as

$$
\begin{aligned}
\mathbb{E}[\square_2] &= \mathbb{E}\left[\frac{1}{n}\sum_{i=1}^{n}\ell_{\mathrm{d}}\big(s^{f,G}(q_i,d_i),s^{F,G}(q_i,d_i)\big) - \frac{1}{n}\sum_{i=1}^{n}\ell_{\mathrm{d}}\big(s^{F,G}(q_i,d_i),s^{F,G}(q_i,d_i)\big)\right] \\
&= \mathbb{E}\left[\big(1-\sigma(F(q)^\top G(d))\big)\,f(q)^\top G(d) + \gamma(-f(q)^\top G(d))\right] \\
&\quad - \mathbb{E}\left[\big(1-\sigma(F(q)^\top G(d))\big)\,F(q)^\top G(d) + \gamma(-F(q)^\top G(d))\right] \\
&= \mathbb{E}\left[G(d)^\top(f(q)-F(q))\big(1-\sigma(F(q)^\top G(d))\big) + \gamma(-f(q)^\top G(d)) - \gamma(-F(q)^\top G(d))\right] \\
&\leq \mathbb{E}\left[\|G(d)\|\|f(q)-F(q)\| + \big|f(q)^\top G(d) - F(q)^\top G(d)\big|\right] \\
&\leq 2K_D\mathbb{E}\left[\|f(q)-F(q)\|_2\right].
\end{aligned}
\tag{17}
$$

$\mathbb{E}[\square_3]$ can be bounded as

$$
\begin{aligned}
\mathbb{E}[\square_3] &= \mathbb{E}[\tilde{R}(F,G) - R(F,G)] \\
&= \mathbb{E}_{(q,d,y)}\left[\ell_{\mathrm{d}}\big(s^{F,G}(q,d),s^{F,G}(q,d)\big) - \ell\big(s^{F,G}(q,d),y\big)\right] \\
&= \mathbb{E}_{(q,d,y)}\left[\big(1-\sigma(F(q)^\top G(d))\big)\,F(q)^\top G(d) + \gamma(-F(q)^\top G(d))\right] \\
&\quad - \mathbb{E}_{(q,d,y)}\left[(1-y)F(q)^\top G(d) + \gamma(-F(q)^\top G(d))\right] \\
&= \mathbb{E}_{(q,d,y)}\left[\big\{1-\sigma(F(q)^\top G(d)) - (1-y)\big\}F(q)^\top G(d)\right] \\
&\leq K_Q K_D\mathbb{E}_{(q,d,y)}\big|\sigma(F(q)^\top G(d)) - y\big|.
\end{aligned}
\tag{18}
$$

Combining (15), (16), (17), and (18) gives the result. $\qquad\square$

## C.2 UNIFORM DEVIATION BOUND

Let $\mathcal{F}$ denote the class of functions that map queries in $\mathcal{Q}$ to their embeddings in $\mathbb{R}^k$ via the query encoder. Define $\mathcal{G}$ analogously for the doc encoder, which consists of functions that map documents in $\mathcal{D}$ to their embeddings in $\mathbb{R}^k$. To simplify exposition, we assume that each training example consists of a single relevant or irrelevant document for each query, i.e., $L=1$ in Section 3. Let

$$
\mathcal{FG} = \{(q,d)\mapsto f(q)^\top g(d) \mid f\in\mathcal{F}, g\in\mathcal{G}\}
$$

Given $\mathcal{S}_n = \{(q_i,d_i,y_i) : i\in[n]\}$, let $N(\epsilon,\mathcal{H})$ denote the $\epsilon$-covering number of a function class $\mathcal{H}$ with respect to $L_2(\mathbb{P}_n)$ norm, where $\|h\|_{L_2(\mathbb{P}_n)}^2 := \|h\|_n^2 := \frac{1}{n}\sum_{i=1}^{n}\|h(q_i,d_i)\|_2^2$. Depending on the context, the functions in $\mathcal{H}$ may map to $\mathbb{R}$ or $\mathbb{R}^d$.

**Proposition 4.** *Let $s^{\mathrm{t}}$ be scorer of a teacher model and $\ell_{\mathrm{d}}$ be a distillation loss function which is $L_{\ell_{\mathrm{d}}}$-Lipschitz in its first argument. Let the embedding functions in $\mathcal{F}$ and $\mathcal{G}$ output vectors with $\ell_2$ norms at most $K$. Define the uniform deviation*

$$\mathcal{E}_n(\mathcal{F}, \mathcal{G}) = \sup_{f \in \mathcal{F}, g \in \mathcal{G}} \frac{1}{n} \sum_{i \in [n]} \ell_{\mathrm{d}}\big(f(q_i)^\top g(d_i), s^{\mathrm{t}}_{q_i, d_i}\big) - \mathbb{E}_{q,d} \ell_{\mathrm{d}}\big(f(q)^\top g(d), s^{\mathrm{t}}_{q,d}\big).$$

*For any $g^* \in \mathcal{G}$, we have*

$$\mathbb{E}_{\mathcal{S}_n} \mathcal{E}_n(\mathcal{F}, \mathcal{G}) \leq \mathbb{E}_{\mathcal{S}_n} \frac{48 K L_{\ell_{\mathrm{d}}}}{\sqrt{n}} \int_0^\infty \sqrt{\log N(u, \mathcal{F}) + \log N(u, \mathcal{G})} \, du,$$

$$\mathbb{E}_{\mathcal{S}_n} \mathcal{E}_n(\mathcal{F}, \{g^*\}) \leq \mathbb{E}_{\mathcal{S}_n} \frac{48 K L_{\ell_{\mathrm{d}}}}{\sqrt{n}} \int_0^\infty \sqrt{\log N(u, \mathcal{F})} \, du.$$

*Proof of Proposition 4.* We first symmetrize excess risk to get Rademacher complexity, then bound the Rademacher complexity with Dudley's entropy integral.

For a training set $\mathcal{S}_n$, the empirical Rademacher complexity of a class of functions $\mathcal{H}$ that maps $\mathcal{Q} \times \mathcal{D}$ to $\mathbb{R}$ is defined by

$$\mathrm{Rad}_n(\mathcal{H}) = \mathbb{E}_\sigma \sup_{h \in \mathcal{H}} \frac{1}{n} \sum_{i=1}^n \varepsilon_i h(q_i, d_i),$$

where $\{\varepsilon_i\}$ denote i.i.d. Rademacher random variables taking the value in $\{+1, -1\}$ with equal probability. By symmetrization (Bousquet et al., 2004) and the fact that $\ell_{\mathrm{d}}$ is $L_{\ell_{\mathrm{d}}}$-Lipschitz in its first argument, we get

$$E_{\mathcal{S}_n} \mathcal{E}_n(\mathcal{F}, \mathcal{G}) \leq 2 L_{\ell_{\mathrm{d}}} \mathbb{E}_{\mathcal{S}_n} \mathrm{Rad}_n(\mathcal{F}\mathcal{G}).$$

Then, Dudley's entropy integral (see, e.g., Ledoux & Talagrand, 1991) gives

$$\mathrm{Rad}_n(\mathcal{F}\mathcal{G}) \leq \frac{12}{\sqrt{n}} \int_0^\infty \sqrt{\log N(u, \mathcal{F}\mathcal{G})} \, du.$$

From Lemma 1 with $K_Q = K_D = K$, for any $u > 0$,

$$N(u, \mathcal{F}\mathcal{G}) \leq N\left(\frac{u}{2K}, \mathcal{F}\right) N\left(\frac{u}{2K}, \mathcal{G}\right).$$

Putting these together,

$$\mathbb{E}_{\mathcal{S}_n} \mathcal{E}_n(\mathcal{F}, \mathcal{G}) \leq \frac{24 L_{\ell_{\mathrm{d}}}}{\sqrt{n}} \int_0^\infty \sqrt{\log N(u/2K, \mathcal{F}) + \log N(u/2K, \mathcal{G})} \, du. \tag{19}$$

Following the same steps with $\mathcal{G}$ replaced by $\{g^*\}$, we get

$$\mathbb{E}_{\mathcal{S}_n} \mathcal{E}_n(\mathcal{F}, \{g^*\}) \leq \frac{24 L_{\ell_{\mathrm{d}}}}{\sqrt{n}} \int_0^\infty \sqrt{\log N(u/2K, \mathcal{F})} \, du \tag{20}$$

By changing variable in Eq. 19 and Eq. 20, we get the stated bounds. □

For $f : \mathcal{Q} \to \mathbb{R}^k, g : \mathcal{D} \to \mathbb{R}^k$, define $fg : \mathcal{Q} \times \mathcal{D} \to \mathbb{R}$ by $fg(q, d) = f(q)^\top g(d)$.

**Lemma 1.** *Let $f_1, \ldots, f_N$ be an $\epsilon$-cover of $\mathcal{F}$ and $g_1, \ldots, g_M$ be an $\epsilon$-cover of $\mathcal{G}$ in $L_2(\mathbb{P}_n)$ norm. Let $\sup_{f \in \mathcal{F}} \sup_{q \in \mathcal{Q}} \|f(q)\|_2 \leq K_Q$ and $\sup_{g \in \mathcal{G}} \sup_{d \in \mathcal{D}} \|g(d)\|_2 \leq K_D$. Then,*

$$\{f_i g_j \mid i \in [N], j \in [M]\}$$

*is a $(K_Q + K_D)\epsilon$-cover of $\mathcal{F}\mathcal{G}$.*

*Proof of Lemma 1.* For arbitrary $f \in \mathcal{F}, g \in \mathcal{G}$, there exist $\tilde{f} \in \{f_1, \ldots, f_N\}, \tilde{g} \in \{g_1, \ldots, g_M\}$ such that $\|f - \tilde{f}\|_n \leq \epsilon, \|g - \tilde{g}\|_n \leq \epsilon$. It is sufficient to show that $\|fg - \tilde{f}\tilde{g}\|_n \leq (K_Q + K_D)\epsilon$. Decomposing using triangle inequality,

$$\|fg - \tilde{f}\tilde{g}\|_n = \|fg - f\tilde{g} + f\tilde{g} - \tilde{f}\tilde{g}\|_n$$
$$\leq \|fg - f\tilde{g}\|_n + \|f\tilde{g} - \tilde{f}\tilde{g}\|_n. \tag{21}$$

To bound the first term, using Cauchy-Schwartz inequality, we can write

$$\frac{1}{n} \sum_{i=1}^{n} \left( f(q_i)^\top g(d_i) - \tilde{f}(q_i)^\top \tilde{g}(d_i) \right)^2 \leq \sup_{q \in \mathcal{Q}} \|f(q)\|_2^2 \cdot \frac{1}{n} \sum_{i=1}^{n} \|(g - \tilde{g})(d_i)\|_2^2.$$

Therefore

$$\|fg - f\tilde{g}\|_n \leq K_Q \|g - \tilde{g}\|_n \leq K_Q \epsilon.$$

Similarly

$$\|f\tilde{g} - \tilde{f}\tilde{g}\|_n \leq K_D \|f - \tilde{f}\|_n \leq K_D \epsilon$$

Plugging these in Eq. 21, we get

$$\|fg - \tilde{f}\tilde{g}\|_n \leq (K_Q + K_D)\epsilon.$$

This completes the proof. □

## D  EVALUATION METRIC DETAILS

For NQ, we evaluate models with full *strict* recall metric, meaning that the model is required to find a *golden* passage from the whole set of candidates (21M). Specifically, for $k \geq 1$, recall@$k$ or R@$k$ denotes the percentage of questions for which the associated golden passage is among the $k$ passages that receive the highest relevance scores by the model. In addition, we also present results for *relaxed* recall metric considered by Karpukhin et al. (2020), where R@$k$ denotes the percentage of questions where the corresponding answer string is present in at least one of the $k$ passages with the highest model (relevance) scores.

For MSMARCO, we follow the standard evaluation metrics *Mean Reciprocal Rank*(MRR)@10 and *normalized Discounted Cumulative Gain* (nDCG)@10, which are computed with respect to BM25 generated 1000 candidate passages for each query. We report $100 \times$ MRR@10 and $100 \times$nDCG@10, as per the convention followed in the prior works.

## E  QUERY GENERATION DETAILS

We introduced query generation to encourage geometric matching in local regions, which can aid in transferring more knowledge in confusing neighborhoods. As expected, this further improves the distillation effectiveness on top of the embedding matching. To focus on the local regions, we generate queries from the observed examples by adding local perturbation in the data manifold (embedding space). Specifically, we employ an off-the-shelf encoder-decoder model (BART). First, we embed an observed query in the corresponding dataset. Second, we add a small perturbation to the query embedding. Finally, we decode the perturbed embedding to generate a new query in the input space. Formally, the generated query $x'$ given an original query $x$ takes the form $x' = \text{Dec}(\text{Enc}(x) + \epsilon)$, where $\text{Enc}()$ and $\text{Dec}()$ correspond to the encoder and the decoder from the off-the-shelf model, respectively, and $\epsilon$ is an isotropic Gaussian noise. Furthermore, we also randomly mask the original query tokens with a small probability. We generate two new queries from an observed query and use them as additional data points during our distillation procedure.

As a comparison, we tried adding the same size of random sampled queries instead of the ones generated via the method described above. That did not show any benefit, which justifies the use of our query/question generation method.

## F  ADDITIONAL TRAINING DETAILS

**Training for teacher models.** For the teacher DE model on NQ, we initialize its question and document encoders by two pre-trained RoBERTa-base models (12 layers). Following (Oğuz et al., 2021), the model is further pre-trained on PAQ (Lewis et al., 2021) for 800K steps, and then fine-tuned on NQ train set with the help of in-batch negatives (Karpukhin et al., 2020) for 40K step.

As for the teacher DE model on MSMARCO, it's known that directly training a DE model on MSMARCO training set leads to poor performance (Menon et al., 2022). Thus, we first train a

`[CLS]`-pooled CE model on triples in MSMARCO training set by using cross-entropy loss. We subsequently use the same triples to distill the resulting CE model to a DE model that has two pre-trained RoBERTa-base models as its two encoders. We utilize cross-entropy based distillation loss in Eq. 10.

**Optimization.** For all of our experiments, we use ADAM weight decay optimizer with a short warm up period and a linear decay schedule. We use the initial learning rate of $10^{-5}$ and $2.8 \times 10^{-5}$ for experiments on NQ and MSMARCO, respectively. We chose batch sizes to be 128.

## G ADDITIONAL EXPERIMENT RESULTS

### G.1 ADDITIONAL EXPERIMENT RESULTS ON NQ

See Table 5 for the performance of various DE models on NQ, as measured by the *relaxed* recall metric.

| Method | Recall@5 | | Recall@20 | | Recall@100 | |
|---|---|---|---|---|---|---|
| | 82M | 16M | 82M | 16M | 82M | 16M |
| Train student directly | 82.0 | 61.7 | 84.4 | 79.9 | 93.8 | 90.6 |
|   + Distill from teacher | 69.8 | 64.6 | 84.5 | 79.7 | 92.4 | 89.2 |
|   + Inherit document embeddings | 76.3 | 67.5 | 88.7 | 81.0 | 94.3 | 89.6 |
|   + Query embedding matching | 80.6 | 75.9 | 91.4 | 88.1 | 96.3 | 94.5 |
|   + Query generation | 80.7 | 77.1 | 91.6 | 89.2 | 96.5 | 94.9 |
| Train student only using embedding matching and inherit doc embeddings | 81.0 | 75.5 | 91.7 | 87.3 | 96.4 | 93.8 |
|   + Query generation | 81.5 | 76.8 | 91.8 | 88.5 | 96.5 | 94.5 |

Table 5: *Relaxed* recall performance of various student DE models on NQ dev set, including symmetric DE student model (82M or 16M transformer for both encoders), and asymmetric DE student model (82M or 16M transformer as query encoder and document embeddings inherited from the teacher). All distilled students used the same teacher (110M parameter RoBERTa-base models as both encoders), with the performance (in terms of relaxed recall) of Recall@5 = 82.5, Recall@20 = 92.6, Recall@100 = 97.1.

### G.2 ADDITIONAL EXPERIMENT RESULTS ON MSMARCO

See Table 6 for CE to DE distillation results on MSMARCO, as measured by the nDCG@10 metric.

| Method | nDCG@10 |
|---|---|
| `[CLS]`-pooled teacher | 43.0 |
| Dual-pooled teacher | 42.8 |
| Standard distillation from `[CLS]`-teacher | 38.8 |
|   +Joint matching | 38.0 |
| Standard distillation from Dual-pooling teacher | 39.2 |
|   +Query matching | 39.4 |

Table 6: Performance of CE to DE distillation on MSMARCO, as measured by the nDCG@10 metric. As for the teacher CE models, we consider two kinds of CE models based on two different pooling mechanism.

## H EMBEDDING ANALYSIS

### H.1 DE TO DE DISTILLATION

Traditional score matching-based distillation might not result in transfer of relative geometry from teacher to student. To assess this, we look at the discrepancy between the teacher and student query embeddings for all $q, q'$ pairs: $\|\mathsf{emb}_q^{\mathsf{t}} - \mathsf{emb}_{q'}^{\mathsf{t}}\| - \|\mathsf{emb}_q^{\mathsf{s}} - \mathsf{emb}_{q'}^{\mathsf{s}}\|$. Note that the analysis is based on NQ, and we focus on the teacher and student DE models based on RoBERTa-base

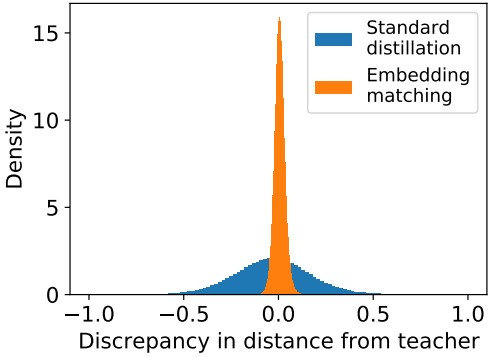

Figure 4: Histogram of teacher-student distance discrepancy in queries.

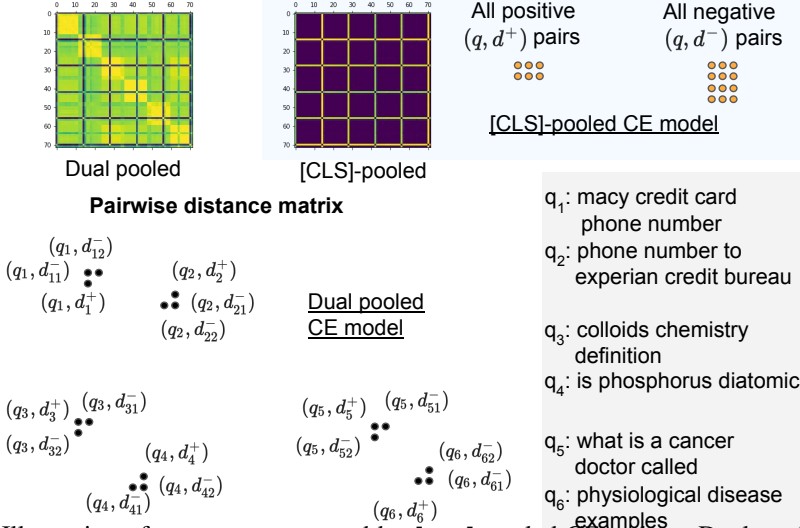

Figure 5: Illustration of geometry expressed by [CLS]-pooled CE and our Dual-pooled CE model on 6 queries from MSMARCO and 12 passages based on pairwise distance matrix across these 72 pairs. [CLS]-pooled CE embeddings degenerates as all positive and negative query-document pairs almost collapse to two points and fail to capture semantic information. In contrast, our Dual-pooled CE model leads to much richer representation that can express semantic information.

and DistilRoBERTa, respectively. As evident from Fig. 4, embedding matching loss significantly reduces this discrepancy.

## H.2    CE TO DE DISTILLATION

We qualitatively look at embeddings from CE model in Fig. 5. The embedding $\mathrm{emb}^{\mathrm{t}}_{q,d}$ from [CLS]-pooled CE model does not capture semantic similarity between query and document as it is solely trained to classify whether the query-document pair is relevant or not. In contrast, the (proxy) query embeddings $\mathrm{emb}^{\mathrm{t}}_{q\leftarrow(q,d)}$ from our Dual-pooled CE model with reconstruction loss do not degenerate and its embeddings groups same query whether conditioned on positive or negative document together. Furthermore, other related queries are closer than unrelated queries. Such informative embedding space would aid distillation to a DE model via embedding matching.

