# OpenReview forum: "EmbedDistill: A geometric knowledge distillation for information retrieval"
_ICLR.cc/2023/Conference — Submitted to ICLR 2023_

### Official Review · Reviewer_VV1L · 2022-10-24

**Confidence:** 3
**Correctness:** 3
**Technical Novelty And Significance:** 3
**Empirical Novelty And Significance:** 2
**Recommendation:** 5

**Clarity, Quality, Novelty And Reproducibility:**

The paper is well-written and has sold theoretical analysis. The idea that aligns the embedding space between teacher and student seems not novel.

**Strength And Weaknesses:**

Strength:
1. The paper is generally well-written and easy to follow.
2. The paper proposes a new idea of distillation approach that goes beyond score matching and aligns the embedding spaces of the teacher and student models.
3. The paper has solid theoretical analysis on the proposed method

Weakness:
1. The question generation method is not well described. As a main contribution emphasized by the authors, more details are required.
2. Both the CE and DE models are based on the RoBERTa-base or RoBERTa-mini, where the model size is no larger than 124M. The authors can try some larger models to show the distillation performance on these larger models.
3. The idea that aligns the embedding space between teacher and student seems not novel.
4. No distillation baselines are compared in the paper.


**Summary Of The Paper:**

The paper proposes a new distillation approach that supports both retrieval and re-ranking stages and crucially leverages the relative geometry among queries and documents learned by the large teacher model. The paper provides stronger signals about local geometry via embedding matching and attains better coverage of data manifold globally via query generation.

**Summary Of The Review:**

Overall, the paper is well-written and easy to follow. But it can be greatly improved after the concern about the technical details on question generation and the experimental settings.

---

> ### Author Response · Authors · 2022-11-15
> **Response to Reviewer VV1L**
>
> Thank you for the time to review our submission. We are happy to learn that the reviewer found that the manuscript was easy to read and conveys a novel idea with solid theoretical analysis. Below, we expand on our query/question generation procedure and aim to address your remaining concerns/comments. We hope that the reviewer will reassess our submission in light of our response.
>
> > ... technical details on question generation ...
>
> We will update the paper to include further details of the query generation as per the following discussion.
>
> We introduced query generation to encourage geometric matching in local regions, which can aid in transferring more knowledge in confusing neighborhoods. As expected, this further improves the distillation effectiveness on top of the embedding matching. To focus on the local regions, we generate queries from the observed examples by adding local perturbation in the data manifold (embedding space). Specifically, we employ an off-the-shelf encoder-decoder model (BART). First, we embed an observed query in the corresponding dataset. Second, we add a small perturbation to the query embedding. Finally, we decode the perturbed embedding to generate a new query in the input space. Formally, the generated query $x’$ given an original query $x$ takes the form $ x’ = Dec( Enc(x) + \epsilon ) $, where $Enc()$ and $Dec()$ correspond to the encoder and the decoder from the off-the-shelf model, respectively, and $\epsilon$ is an isotropic gaussian noise. Furthermore, we also randomly mask the original query tokens with a small probability. We generate two new queries from an observed query and use them as additional data points during our distillation procedure.
>
> As a comparison, we tried adding the same size of random sampled queries instead of the ones generated via the method described above. That did not show any benefit, which justifies the use of our query/question generation method.
>
> > ... can try some larger models ...
>
> Per reviewer’s suggestion, we just started conducting similar experiments with RoBERTa-Large teacher and will share the results once they are available.
>
> At the same time, we would like to mention that 12-layer models (like RoBERTa-Base) have been frequently employed in the recent literature as teacher models (see, e.g., Table 3, [1], and [2]).
>
> > ... idea that aligns the embedding space between teacher and student ...
>
> We agree that embedding alignment between a teacher and a student itself is not novel, as we also discussed in Section 2 under the heading “Distillation with representation alignments”.
>
> Instead, our novelty is in introducing a novel distillation algorithm in the IR setting where we have DE/CE models. We further provide theoretical insight on why this proposed algorithm can achieve better performance.
>
> > ... distillation baselines ...
>
> We did extensive search to provide more baselines particularly DE->DE distillation with similar model capacity but we could not find any precedent literature. (We will be happy to cite any work which the reviewer is aware of.) It is due that we are exploring much smaller model sizes (e.g., mini size model with 4-layer transformer with 256 dimension) and **asymmetric DE architecture** inheriting the teacher’s DE tower.
>
> Hence, we tried to provide an extensive list of ablation (e.g., Table 1) to demonstrate how each factors such as using the asymmetric architecture or using EmbedDistil with or without the traditional distillation objective and also with or without the query generation (they are still largely based on distillation). What we tried to demonstrate here is that the new training objective, i.e., query embedding matching, was very effective and we were able to achieve 99% of performance using 1/2-sized query encoder and 94% using 1/8-sized query encoder. In terms of FLOPs count, the latter corresponds to 1.4G FLOPs (only 1/15 of 22G FLOPs of the teacher) while maintaining 91%+ of the performance.
>
> Table 3 also provides related baselines in a wider scope. This proposed mini model (4 layer with 256 dimension) performs very close to the original DPR paper model 76.1 vs 78.4 albeit it uses 1/8 of the query encoder size.
>
> [1] Hofstatter et al., *Improving efficient neural ranking models with cross-architecture knowledge distillation*. https://arxiv.org/abs/2010.02666
>
> [2] Menon et al., *In defense of dual-encoders for neural ranking*, Proceedings of the 39th International Conference on Machine Learning, PMLR 162:15376-15400, 2022.

---

> ### Author Response · Authors · 2022-12-06
> **Additional Results with the Large Teacher**
>
> Here’re the results that the reviewers requested with the larger teacher model (Roberta Large -- 24 layer model).
>
> RoBERTa Large -> 82M model
>
> | Method                             | Recall@5 | Recall@20 | Recall@100 |
> | ---------------------------------- | -------- | --------- | ---------- |
> | Teacher                            |   68.2   |   84.1    |   92.1     |
> |  |  |  |  |
> | Train student directly             |   41.9   |   64.5    |   82.0     |
> | + Distill from teacher             |   58.3   |   75.5    |   86.2     |
> | + Inherit document embeddings      |   61.6   |   78.3    |   88.5     |
> | + Query embedding matching         |   63.7   |   80.8    |   90.7     |
> | + Query generation                 |   64.5   |   81.5    | **91.2**   |
> |  |  |  |  |
> | Train student only using embedding matching and inherit doc embeddings|   66.2   |   81.3    |   90.6     |
> | + Query generation                 | **66.8** | **82.1**  | **91.0**   |
>
> RoBERTa Large -> 16M model
>
> | Method                             | Recall@5 | Recall@20 | Recall@100 |
> | ---------------------------------- | -------- | --------- | ---------- |
> | Teacher                            |   68.2   |   84.1    |   92.1     |
> |  |  |  |  |
> | Train student directly             |   39.5   |   59.9    |   76.3     |
> | + Distill from teacher             |   43.4   |   59.3    |   73.2     |
> | + Inherit document embeddings      |   45.8   |   61.4    |   73.8     |
> | + Query embedding matching         |   57.3   |   73.7    |   85.4     |
> | + Query generation                 | **58.2** | **75.1**  | **86.3**   |
> |  |  |  |  |
> | Train student only using embedding matching and inherit doc embeddings |   58.8   |   74.9    |   85.2     |
> | + Query generation                 |   57.8   |   74.9    |   85.4     |
>
>
> As expected, the performance has significantly improved for the 6-layer model (82M model) -- see the RoBERTa-base teacher result from Table 1 (recall@20 was 80.5 from our best model).
>
> Similar to the previous result, we were able to achieve 97% (recall@20) of the teacher’s performance with only a fraction of the model size. Please also note that this (recall@20=82.1) is even better than the RoBERTa-base teacher that had  recall@20 of 81.7.
>
> Unlike the 6-layer model, the 4-layer model did not improve the performance much probably due to the larger teacher-student **capacity** gap [1].
>
> [1] Jang Hyun Cho and Bharath Hariharan. On the efficacy of knowledge distillation. In Proceedings of the IEEE/CVF international conference on computer vision, pp. 4794–4802, 2019.

---

### Official Review · Reviewer_CgJL · 2022-10-24

**Confidence:** 4
**Correctness:** 4
**Technical Novelty And Significance:** 3
**Empirical Novelty And Significance:** 3
**Recommendation:** 5

**Clarity, Quality, Novelty And Reproducibility:**

Clarify: can be improved.

Quality: good

Novelty: fair, The embedDistill is similar to the feature-based distillation. There are many previous works.
Reproducibility: good

**Strength And Weaknesses:**

Strength:

     -- The theory proof makes the method more convincing.
     -- Try to distill the knowledge based on the embeddings(or, say, outputs) from Transformer.


Weakness:

    -- Poor writing and too many long sentences in the Introduction Section.
    -- Lack of comparison. The experiments seem poor, and they only compare the proposed embedDistill with naïve logit-based distillation, and they do not compare with other methods mentioned in related work.

**Summary Of The Paper:**

This paper proposes a novel embedding-based knowledge distillation method for Transformer in QA, including DE2DE and CE2DE. Moreover, they give a theory that proves the effectiveness of the distillation method on the generalization of student networks.

**Summary Of The Review:**

It is a valuable topic to perform feature-based distillation for transformer in QA tasks. But more comparisons are needed to prove the effectiveness, and the writing can be improved. Moreover, embedDistill brings an extra aligned projection matrix, which may take more training resources.

---

> ### Author Response · Authors · 2022-11-15
> **Response to Reviewer CgJL**
>
> We thank the reviewer for reviewing our manuscript. We are glad that the reviewer acknowledged the novelty of our method in the IR setting and liked our theoretical contributions.
>
> > ... too many long sentences.
>
> We will polish the writing to make it easier to follow (especially, splitting long sentences into shorter ones as you suggested) while preparing the final version.
>
> > ... compare with other methods ...
>
> We did extensive search to provide more baselines particularly DE->DE distillation with similar model capacity but we could not find any precedent literature. (We will be happy to cite any work which the reviewer is aware of.) It is due that we are exploring much smaller model sizes (e.g. mini size model with 4-layer transformer with 256 dimension) and asymmetric DE architecture inheriting the teacher’s DE tower. Hence, we tried to provide an extensive list of ablation (e.g. Table 1) to demonstrate how each factors such as using the asymmetric architecture or using EmbedDistil with or without the traditional distillation objective and also with or without the query generation. What we tried to demonstrate here is that the new training objective, i.e., query embedding task, was very effective and we were able to achieve 99% of performance using 1/2-sized query encoder and 94% of performance using 1/8-sized query encoder. In terms of FLOPs count, the latter corresponds to 1.4G FLOPs (only 1/15 of 22G FLOPs of the teacher) while maintaining 91%+ of the performance.
>
> Table 3 also provides related baselines in a wider scope. This proposed mini model (4 layer with 256 dimension) performs very close to the original DPR paper model 76.1 vs 78.4 albeit it uses 1/8 of the query encoder size.
>
> > ... brings an extra aligned projection matrix...
>
> Please note the projection matrix is a lot lighter computationally than the transformer blocks, which we reduced significantly without losing much of the teacher’s performance. For example in NQ dataset (Table 1), we were able to reduce the 12-layer 768-dim query encoder to 6-layer 768-dim encoder (1/2 size) maintaining 99% of the performance, and to 4-layer 256-dim encoder (1/8 size) maintaining 91%+ of the performance. Compared to significant compute savings of compact transformer stacks, additional compute cost of the MLP projection is negligible in practice.
>
> > It is a valuable topic to perform feature-based distillation for transformer in QA tasks.
>
> We agree with the reviewer about the similarity between EmbedDistil and feature-based distillation, as already acknowledged in Section 2. We will expand the relevant discussion, particularly the paragraph on *Distillation with representation alignments*, to have a more comprehensive account of feature-based distillation. Please also note that we do not claim to be the first ones to introduce the distillation with representation alignments. Our contribution focuses on proposing novel training objectives and architectures for the IR setting. Note that our proposals are well supported by accompanying theoretical analysis.

---

> ### Author Response · Authors · 2022-12-06
> **Follow-up**
>
> We again thank the reviewer for their valuable feedback. We have posted a point-by-point response to the reviewer's comment. We would be happy to answer/clarify any comments/concerns the reviewer might have after reading our response. Here we would also like to bring the reviewer's attention to the additional experiments we conducted with a larger 24-layer RoBERTa teacher based on Reviewer VV1L's suggestion.

---

### Official Review · Reviewer_mS5Z · 2022-10-24

**Confidence:** 4
**Correctness:** 2
**Technical Novelty And Significance:** 2
**Empirical Novelty And Significance:** 2
**Recommendation:** 3

**Clarity, Quality, Novelty And Reproducibility:**

The paper is overall clear while lacking novelty. It seems no code is available at this moment and I recommend the authors consider open-sourcing the code.

**Strength And Weaknesses:**

## Strength
1. This paper provides a theoretical justification of the proposed method.
1. This paper includes a good analysis of the embeddings, which empirically justifies their approach.
1. The ablation study (Table 2) is informative.

## Weaknesses
1. The mathematical expressions seem to be overused. There are a lot of unnecessary formulas in the paper, e.g., Eq. 1,2,3
1. The proposed method is not new as there have been many works on distillation for retrieval, both CE to DE and DE to DE.
1. There are mentions of prior studies on text retrieval + knowledge distillation. However, they are not compared in the experiments.
1. I hope to see evaluation on more datasets, e.g., BEIR benchmark.
1. The performance of the student model is somewhat disappointing. It's maybe because of the weak teacher. Why not use a SOTA model as the teacher?

#### Minor issues:
1. Figure 1 is too small.

**Summary Of The Paper:**

The authors propose a distillation approach that supports both retrieval and re-ranking stages and crucially leverages the relative geometry among queries and documents learned by the large teacher model.

**Summary Of The Review:**

This paper explores two ways of distilling into dual-encoder text retrieval models. However, it lacks novelty and comparison with other approaches.

---

> ### Author Response · Authors · 2022-11-15
> **Response to Reviewer mS5Z Part 2**
>
> > BEIR benchmark.
>
> Please note that the NQ and MSMARCO datasets that we used are already part of the BEIR benchmark. These two benchmarks are popular for supervised IR settings, whereas other datasets in BEIR are more geared towards zero-shot evaluation, which is not the scope of this paper.
>
> > ... because of the weak teacher.
>
> We tried our best to produce a convincing teacher (for example, we followed [1] to have additional PAQ pretraining before the actual NQ training); however, it was very hard to reproduce the exact number from the existing papers because many of them were based on **multi-staged training**. In case of single stage training, we do not find publicly available models e.g. RoBERTa based retriever on PAQ https://github.com/facebookresearch/dpr-scale is not available. Hence we based our evaluation on our best effort teacher and provided extensive ablations to demonstrate fine-grained understanding of our method.
>
> We wanted to point out that our focus is to compare the different distillation strategies such as standard softmax or inheriting the frozen towers or with/without the query generation given a limited budget in compute. We were able to reproduce 91%+ performance only using a 1/8-sized query encoder (1/15 in inference speed in terms of FLOPs). Also, note that as discussed in the paragraph on Distillation for IR in Sec 2, our work only focuses on single-stage distillation.
>
> > Figure 1 is too small.
>
> Thank you for the feedback. We will increase the size in the final version of the paper.
>
> > ... consider open-sourcing the code.
>
> Thank you for your suggestion regarding open sourcing the codebase. We certainly plan to do that with the necessary approval process, when the paper is accepted.
>
> [1] Oğuz, Barlas, et al. "Domain-matched pre-training tasks for dense retrieval." arXiv preprint arXiv:2107.13602 (2021).

---

> > ### Comment · Reviewer_mS5Z · 2022-11-23
> > **Teacher Model**
> >
> > I'm not sure why it's not possible to have a stronger teacher. For example, a teacher trained with multi-stage/hard negatives mining doesn't conflict with a single-stage distillation. In Table 3, there lacks a model that has a similar runtime as the proposed one. One possible reason is the weak teacher and I would like to see what's the limit of the proposed method if distilled from a SOTA model. Thus, the evaluation is not quite unconvincing to me. I still think the authors should try harder to address this in the revision.

---

> > > ### Author Response · Authors · 2022-12-06
> > > **Re: Teacher model**
> > >
> > > Thank you for your comment. Indeed, a teacher trained with multi-stage/hard negative mining does not conflict with our single-stage distillation. Since utilizing a stronger teacher is orthogonal to our proposal, it can further improve the student’s performance. However, (as we mentioned in our earlier response,) given the difficulty of reproducing such strong teachers in our codebase  and the limited time, we instead followed an approach hinted by Reviewer VV1L regarding utilizing a stronger teacher.
> > >
> > > We evaluated our proposed distillation method on NQ dataset with a much larger (and better performing) teacher model based on the **24 layer** RoBERTa-Large architecture (note that we had used the **12 layer** RoBERTa-base architecture in our existing experiments). It improved our results further as expected and even the student model outperformed the RoBERTa-based teacher (recall@20 was 81.7 from the previous teacher).
> > >
> > > RoBERTa Large -> 82M model
> > >
> > > | Method                             | Recall@5 | Recall@20 | Recall@100 |
> > > | - | - | - | - |
> > > | Teacher                            |   68.2   |   84.1    |   92.1     |
> > > |   |   |   |   |
> > > | Train student directly             |   41.9   |   64.5    |   82.0     |
> > > | + Distill from teacher             |   58.3   |   75.5    |   86.2     |
> > > | + Inherit document embeddings      |   61.6   |   78.3    |   88.5     |
> > > | + Query embedding matching         |   63.7   |   80.8    |   90.7     |
> > > | + Query generation                 |   64.5   |   81.5    | **91.2**   |
> > > |   |   |   |   |
> > > | Train student only using embedding matching and inherit doc embeddings |   66.2   |   81.3    |   90.6     |
> > > | + Query generation                 | **66.8** | **82.1**  | **91.0**   |

---

> ### Author Response · Authors · 2022-11-15
> **Response to Reviewer mS5Z Part 1**
>
> We appreciate your effort in reviewing our submission. We are glad that the reviewer found our theoretical analysis and empirical evaluations (especially the ablation studies) valuable. Below, we aim to address the reviewer's concerns and hope the reviewer can reconsider their assessment of this paper.
>
> > The paper is overall clear while lacking novelty.
>
> First of all, we would like to reiterate our main contributions. Our contributions lie in providing theoretical understandings of IR model distillation, providing novel architectures/settings (e.g., asymmetric DE with frozen document encoder), and proposing novel efficient learning algorithms (embedding matching and query generation) in the context of IR distillation. We do not claim that we are the first ones to explore embedding alignments for neural networks (see the discussion in Section 2 under the heading “Distillation with representation alignments.”)
>
> > The mathematical expressions seem to be overused
>
> We have tried to be rigorous in our exposition. Please note that the terms defined in Eq. 1, 2, and 3 are also critical to understand the theoretical findings in Sec. 5. That said, if the reviewer feels strongly about this point, we will simplify Section 3 and lighten up the use of mathematical notations. For example, we can move Eq. 1, 2, and 3 to the appendix.
>
> > ... not compared in the experiments.
>
> We did extensive search to provide more baselines particularly DE->DE distillation with similar model capacity but we could not find any precedent literature. (We will be happy to cite any work which the reviewer is aware of.) It is due that we are exploring much smaller model sizes (e.g. mini size model with 4-layer transformer with 256 dimension) and asymmetric DE architecture inheriting the teacher’s DE tower. Hence, we tried to provide an extensive list of ablation (e.g. Table 1) to demonstrate how each factors such as a) using the asymmetric architecture, b) using EmbedDistil with or without the traditional distillation objective, and c) using query generation during embedding matching task. What we tried to demonstrate here is that the new training objective, i.e., query embedding matching, was very effective and we were able to achieve 99% of performance using 1/2-sized query encoder and 94% using 1/8-sized query encoder. In terms of FLOPs count, the latter corresponds to 1.4G FLOPs (only 1/15 of 22G FLOPs of the teacher) while maintaining 91%+ of the performance.
>
> Table 3 also provides related baselines in a wider scope. This proposed mini model (4 layer with 256 dimension) performs very close to the original DPR paper model 76.1 vs 78.4 while it uses 1/8-sized query encoder.

---

> > ### Comment · Reviewer_mS5Z · 2022-11-23
> > **Baselines**
> >
> > I know it's not an apple-to-apple comparison, but what about a paper like this one: https://arxiv.org/abs/2104.06967
> >
> > It would be great to compare to KD methods, although the setting is not entirely the same.

---

> > > ### Author Response · Authors · 2022-12-06
> > > **Re: Baselines**
> > >
> > > We thank the reviewer for pointing us to [1]. However, as mentioned by the reviewer the setting considered in [1] is different from our main focus, i.e., the dual encoder (DE) to DE distillation. In contrast, [1] studies the settings where one is interested in distilling a DE student model from a cross-encoder (CE) teacher, ColBERT teacher, or dual teacher supervision based on CE and ColBERT models. Furthermore, the main contribution of [1] is to design a training-batch construction scheme, namely **balanced topic aware sampling (TAS-Balanced)**. TAS-Balanced is orthogonal to our approach of modifying the distillation objective and can potentially be combined with our proposed distillation method. We believe that such a study, although interesting, is outside the scope of this submission.
> > >
> > > Here, we would also like to make the following comments regarding **other** (i.e., non-TAS-Balanced) KD methods discussed in [1]:
> > >
> > > * On MSMARCO, our best performing **6-layer** student based on **DE to DE** distillation (MRR@10 = 33.00 in our Table 2) outperforms the best performing **6-layer** student model realized by any non-TAS-balanced method in Table 6 of [1] (MRR@10 = 32.6 by Margin-MSE).
> > >
> > > * Similarly, our best performing **12-layer** student based on **CE to DE**  distillation achieves a higher MRR@10 value of 33.7 in Table 4 (of our submission). This value of MRR@10 is even higher than the MRR@10 value of 33.5 that **12-layer** student in Table 6 [1] achieves.
> > >
> > > [1] Hofstätter, Sebastian, et al. "Efficiently teaching an effective dense retriever with balanced topic aware sampling." Proceedings of the 44th International ACM SIGIR Conference on Research and Development in Information Retrieval. 2021.

---

### Official Review · Reviewer_zx15 · 2022-10-25

**Confidence:** 3
**Correctness:** 3
**Technical Novelty And Significance:** 2
**Empirical Novelty And Significance:** 3
**Recommendation:** 6

**Clarity, Quality, Novelty And Reproducibility:**

The paper is mostly clear, and most details for reproducibility are given in the appendix. The novelty of the paper is to apply embedding alignment on top of the distillation loss and to show that this works in IR. Note that embeddings alignment has already been studied (as discussed by the authors), so the novelty is on how to apply it to an IR setting where there are small differences (two representations instead of one, and the fact that cross-encoders do not have a separate representation for documents and queries).


There are just a few unclear/to correct parts:
- It is not clear how the teacher and student performance are computed: is it on the same re-reranking task (which is not the primary use case of dense retrievers) or on different tasks (re-ranking for the teacher and full ranking for the student).
- the discussion about the fact that the cross-encoder representation cannot be used for aligning document/query representation (sec 4.2) is quite obvious since the CLS token is only trained on separating relevant from non-relevant (query, document) pairs.



**Strength And Weaknesses:**

This paper shows that aligning embeddings (with an ad-hoc procedure in the case of cross-encoders where queries and documents do not have a dense representation) helps improve the performance of student models when distilling.

The various experiments conducted clearly show that the proposed methodology do help in a variety of settings (dense to dense, cross-encoder to dense, dense to lighter dense) with two representative datasets (MS Marco and Natural Questions), showing the robustness of the approach.

The paper provides two propositions justifying the approach (I did not check the proofs in the appendix, but they seem to be sensible), but I failed to understand how they help showing in theory how aligning the embeddings does help improving the student:

1) Proposition 1 shows that the difference between the distillation loss and the teacher loss is bounded by the expectation of embeddings distance and a dense teacher loss: how this shows that minimizing the proposed loss (distillation loss + embeddings loss) will help improving generalization?

2) Proposition 2 states that the difference between the empirical and theoretical loss is bounded by some quantity that is lower when one part of the network is frozen (i.e. the document embeddings part in this paper). If this is the case, first, it seems that just proving the bound would be enough (which should be doable since the Rademacher complexity of the sum of two classes of functions is the sum of their complexities, by decomposing the query representation as $q + \delta_q$ and the same for the document); after that, I wonder what is the practical or theoretical value of such bounds in this paper's context.

I might add that from a theoretical point of view, it would have been interesting to understand (1) what is the last term of the equation of proposition 1 and how it relate to the performance of the model (2) to study the difference between the "[CLS]" classification and the one based on the dense embeddings obtained through pooling.


**Summary Of The Paper:**

This paper proposes a new distillation technique for learning Information Retrieval (IR) models whereby not only the scores of the documents are aligned with the teacher, but also the representations. The paper exhibits two propositions showing (1) a relationship involving the empirical distillation loss and the representations (2) the bias between the empirical and expected distillation loss. Experiments show that the regularization is based on embeddings. The authors show that this improves the result with two types of distillation (dense to dense, and cross-encoder to dense).


**Summary Of The Review:**

This paper proposes to regularize the distillation loss by aligning the query and document representations (or a substitute when they are not readily available, e.g. for cross-encoders), and clearly shows its benefit on two standard IR datasets. The theoretical section fails to show anything (IMHO) that could really be leveraged or provide insight on what this distillation is doing, and this lessen the interest of the paper (a qualitative or more focused theoretical analysis would have been more valuable).

---

> ### Author Response · Authors · 2022-11-15
> **Response to Reviewer zx15 Part 2**
>
> > Proposition 2 ...
>
> We agree with the reviewer’s high-level interpretation of Proposition 2 that the difference between the empirical and theoretical loss is bounded by some quantity that is lower when one part of the network is frozen. However, we would like to highlight that formally proving this is not straightforward even if it feels very intuitive. Unlike in the case of sum of two networks, our IR models are based on inner products for which Rademacher complexity does not decompose into sum of individual complexities. In the worst case, depending on exact definition function classes, in fact freezing one network might not even have any reduction in Rademacher complexity, (e.g. if one of the networks is complex enough). We can at best hope to obtain a tighter upper bound for the frozen case than both networks being trainable. In this regard, we first derive Lemma 1 (Appendix C) where we leverage the special inner-product based architecture of our model to express the relationship between covering numbers of individual function classes and our inner-product based function class. Then this relationship can be leveraged to compare the Rademacher complexity of one frozen network model vs both trainable network models.
>
> > ... what is the last term of the equation of proposition 1 ...
>
> As mentioned above, the third term in Proposition 1 captures how well the teacher model predicts the true relevance label y. Since the teacher is fixed during the distillation, this term does not help differentiate the performance of different students.
>
> > ... study the difference between the "[CLS]" classification and the one based on the dense embeddings obtained through pooling.
>
> We note that the difference between the [CLS] and mean pooling has broader significance beyond just IR models, which is out of the scope of this paper. Thus, we feel that the reviewer’s suggestion is an interesting avenue for future work.
>
> > It is not clear how the teacher and student performance are computed ...
>
> We will update the paper to clarify evaluation metrics. For NQ, we rely on the full recall evaluation for the dev set, or the relaxed recall evaluation for the test set (similar to full recall, but with a relaxed matching following [1]). For MSMARCO, we conduct the re-ranking evaluation using standard MRR@10.
>
> > The discussion about the fact that the cross-encoder representation cannot be used ...
>
> We thought it might not be very apparent for some readers. We may shorten this discussion to save some space for other sections.
>
> [1] Karpukhin, Vladimir, et al. "Dense passage retrieval for open-domain question answering." arXiv preprint arXiv:2004.04906 (2020).

---

> ### Author Response · Authors · 2022-11-15
> **Response to Reviewer zx15 Part 1**
>
> Thank you for reviewing our submission. We are glad that the reviewer recognized the novelty of our proposal in the IR setting and the wide-spectrum of our empirical evaluations. In the response below, we aim to provide the connection between our theoretical analysis and proposed methods, and address your remaining concerns/comments.
>
> > ... theoretical section fails to show anything (IMHO) that could really be leveraged ...
>
> In IR distillation, we often distill a large DE teacher to a DE student. In this setting, we can bound the teacher-student performance gap using three terms (Proposition 1): a) The expected gap of document embeddings between the teacher and the student b) the expected gap between the query embeddings, and c) how good the teacher is. __Because c) does not depend on the student__, we only focus on a) and b) (Note that c) captures how good *the teacher* is in predicting the true relevance label y). Moreover, when the student inherits the teacher’s document tower (*asymmetric DE architecture*), the bound minimization is equivalent to what we proposed in this paper – the embedding matching loss. In other words, our proposed method is a direct minimization of the theoretical teacher-student gab bounds; hence it performs well.
>
> Based on this theoretical analysis, we get the insight that minimizing these embedding matching should suffice. Indeed, we observed empirically that training the student model via **only** query embedding matching (without the traditional score matching) and inheriting the teacher document encoder consistently outperforms other approaches (see e.g., the last two lines of Table 1 and 2).
>
> Please see below for further details and we will update the paper with further clarifications.
>
> > Proposition 1 shows ...
>
> Proposition 1 does not measure the generalization bound (for the student). It measures the *expected performance gap* between the teacher and the student.
>
> The left-hand side is the difference between the student’s expected empirical risk (first term $R(s^\mathrm{s}, s^\mathrm{t}; \mathcal{S}\_n)$), and the teacher’s population risk (second term $\mathbb{E} \ell(s^t\_{q,d}, y)$). Basically the left side measures the expected performance gap between the teacher and the student.
>
> The right hand side states the left hand side (the performance gap) is upper bounded by some constant factors of the following three quantities. a) RHS first term $\mathbb{E}\_d[||\mathrm{emb}^s\_d - \mathrm{emb}^t\_d ||]$ is the expected difference between the document embeddings of the student and the teacher b) RHS second term $\mathbb{E}\_d[||\mathrm{emb}^s\_q - \mathrm{emb}^t\_q ||]$ is the expected difference between the query embeddings of the student and the student c) RHS third term $\mathbb{E}\_{(q,d,y)} \big|\mathrm{sigmoid}(\langle \mathrm{emb}^t\_q, \mathrm{emb}^t\_d \rangle) -y\big|$ measures the teacher’s performance -- how much the teacher is close at modeling the true probability. Since the teacher is given and fixed, the third term is an irreducible constant. This third term is not a function of the student.
>
> Our proposed EmbedDistill uses $\mathrm{Obj} := R(s^\mathrm{s}, s^\mathrm{t}; \mathcal{S}\_n)  + O\_q + O\_d$  as the objective, where $O\_q := \frac{1}{n}\sum \nolimits_{(q, d) \in \mathcal{S}\_n} \|\|   \mathrm{emb}^t\_q - {\rm proj}\big( \mathrm{emb}^s\_q\big) \|\|$ and $O\_d := \frac{1}{n}\sum \nolimits\_{(q, d) \in \mathcal{S}\_n} \|\|   \mathrm{emb}^t\_d - {\rm proj}\big( \mathrm{emb}^s\_d\big) \|\|$.
>
> Minimizing $\mathrm{Obj}$ thus corresponds to directly making the upper bound on the right-hand side in Proposition 1 tighter. That is, adding the embedding alignment loss helps to reduce the performance gap of the student and the teacher. Further in the setting where the student inherits the teacher’s doc encoder (see Sec 4.1), the bound only relies on the query embedding differences, which is the exact form of the query embedding matching. Hence, our training objective is explicitly related to closing the (expected) performance gap between the teacher and the student.

---

> > ### Comment · Reviewer_zx15 · 2022-11-21
> > **Proposition 1**
> >
> > Thanks for your answers, but I still need some clarification on proposition (1).  Leaving aside the fact that matching the losses does not guarantee the performance of the model, I have two observations.
> >
> > First, it is not obvious that
> >
> > $$
> > \mathbb{E} R(s^s, s^t,; \mathcal S_n) \ge \mathbb{E}(l(s_{q,d}^t,y))
> > $$
> >
> > and if this is not the case, the bound is quite useless.
> >
> > Second, the bound is not tight since if the teacher and student are the exactly same, then the difference in losses is equal to the teacher's loss.
> >
> > Based on both statements, I still fail to see much value in proposition (1) - even though I agree that the idea is appealing intuitively and works in practice.

---

> > > ### Author Response · Authors · 2022-11-22
> > > **Re: Proposition 1**
> > >
> > > We thank the reviewer for their interest in ascertaining the utility of Proposition 1 and raising valid questions in the process. Please see our response below regarding your two observations.
> > >
> > >
> > > > First, it is not obvious that $\mathbb{E} R(s^s, s^t,; \mathcal S_n) \ge \mathbb{E}(l(s_{q,d}^t,y))$ and if this is not the case, the bound is quite useless.
> > >
> > > Please note that it is not guaranteed that whether $\mathbb{E} R(s^s, s^t,; \mathcal S_n) \ge \mathbb{E}l(s_{q,d}^t,y)$ or $\mathbb{E} R(s^s, s^t,; \mathcal S_n) < \mathbb{E}l(s_{q,d}^t,y)$ **always** holds. Depending on the teacher model, student model, and underlying data distribution and samples, the inequality can hold in either direction. That said, the bound in Proposition 1 always holds; and, as pointed out by the reviewer, is particularly interesting when $\mathbb{E} R(s^s, s^t,; \mathcal S_n) < \mathbb{E}l(s_{q,d}^t,y)$.
> > >
> > > Here, we would like to note that the right hand side expression in Proposition 1 even bounds the **absolute** difference $\vert\mathbb{E} R(s^s, s^t,; \mathcal S_n) - \mathbb{E}l(s_{q,d}^t,y)\vert$. (It’s straightforward to verify from the proof of the proposition in Appendix C.1.)
> > >
> > > > Second, the bound is not tight since if the teacher and student are the exactly same, then the difference in losses is equal to the teacher's loss.
> > >
> > > The reviewer is right in pointing out that the bound is still non-zero even when the teacher and student are exactly the same. This can be attributed to the fact that we are measuring the gap between two different kinds of loss functions: 1) **Distillation loss** $R(s^s, s^t,; \mathcal S_n)$ for the student, which is defined wrt. teacher provided supervision; and 2) **Standard loss** $l(s_{q,d}^t,y)$ for the teacher, which is defined wrt. **original** labels $\{y\}$.
> > >
> > > If we had bounded the difference between the **identical** loss, e.g.,
> > >
> > > $$
> > > \mathbb{E}R(s^s, s^t,; \mathcal S_n) - \mathbb{E}R(s^t, s^t,; \mathcal (x, y)),
> > > $$
> > >
> > > the resulting bound would become zero, when the teacher and student are exactly the same. (In this case, we will only have to deal with $\square_{1}$ and $\square_{2}$ in Eq. (15) in Appendix C.1. Thus, the third term in the right hand side of the bound in Proposition 1 would not appear.
> > >
> > > However, $\mathbb{E}R(s^t, s^t,; \mathcal (x, y))$ is not a standard proxy for the teacher model’s performance. One typically employs a **surrogate loss** (wrt. original labels \{y\}) to bound a model’s performance. This is why we focus on $l(s_{q,d}^t,y)$ to capture the teacher model’s performance.
> > >
> > > Here, we would like to emphasize that the purpose of Proposition 1 is to highlight (and justify) the utility of **embedding matching** to reduce the gap between the student and teacher model’s behavior. The bound in Proposition 1 is exactly able to achieve that. One could use the standard loss or distillation loss as a proxy for the teacher’s performance. The bounds resulting from these choices only differ in the terms that **do not** depend on the student models (as a result, these terms don’t play a role in differentiating two student models).

---

### Decision · Program_Chairs · 2023-01-20

**Decision:**

Reject

**Justification For Why Not Higher Score:**

Not good enough

**Justification For Why Not Lower Score:**

N/A

**Metareview: Summary, Strengths And Weaknesses:**

The proposed distillation technique with embedding matching for learning neural IR models in different scenarios (DE to DE and CE to DE) and the theoretical justification are interesting. However, the reviewers were not convinced even after the rebuttal on the: (a) novelty, (b) usefulness of the theoretical proof for distillation, and (c) completeness of experiments/evaluation (e.g., not using a strong teacher, other baselines not compared). I am in favour of rejecting the paper in its current form.



**Summary Of Ac-Reviewer Meeting:**

N/A